# Disulfide bridge-dependent dimerization triggers FGF2 membrane translocation into the extracellular space

Fabio Lolicato[1,2]*[†], Julia P Steringer[1][†], Roberto Saleppico[1], Daniel Beyer[1], Jaime Fernandez-Sobaberas[1], Sebastian Unger[1], Steffen Klein[3], Petra Riegerová[4], Sabine Wegehingel[1], Hans-Michael Müller[1], Xiao J Schmitt[5], Shreyas Kaptan[2], Christian Freund[5], Martin Hof[4], Radek Šachl[4], Petr Chlanda[3], Ilpo Vattulainen[2], Walter Nickel[1]*

[1]Heidelberg University Biochemistry Center, Heidelberg, Germany; [2]Department of Physics, University of Helsinki, Helsinki, Finland; [3]Schaller Research Group, Department of Infectious Diseases-Virology, Heidelberg University Hospital, Heidelberg, Germany; [4]J. Heyrovský Institute of Physical Chemistry of the Czech Academy of Sciences, Prague, Czech Republic; [5]Institute for Chemistry and Biochemistry, Freie Universität Berlin, Berlin, Germany

*For correspondence:
fabio.lolicato@bzh.uni-heidelberg.de (FL);
walter.nickel@bzh.uni-heidelberg.de (WN)

[†]These authors contributed equally to this work

**Abstract** Fibroblast growth factor 2 (FGF2) exits cells by direct translocation across the plasma membrane, a type I pathway of unconventional protein secretion. This process is initiated by phosphatidylinositol-4,5-bisphosphate ($PI(4,5)P_2$)-dependent formation of highly dynamic FGF2 oligomers at the inner plasma membrane leaflet, inducing the formation of lipidic membrane pores. Cell surface heparan sulfate chains linked to glypican-1 (GPC1) capture FGF2 at the outer plasma membrane leaflet, completing FGF2 membrane translocation into the extracellular space. While the basic steps of this pathway are well understood, the molecular mechanism by which FGF2 oligomerizes on membrane surfaces remains unclear. In the current study, we demonstrate the initial step of this process to depend on C95-C95 disulfide-bridge-mediated FGF2 dimerization on membrane surfaces, producing the building blocks for higher FGF2 oligomers that drive the formation of membrane pores. We find FGF2 with a C95A substitution to be defective in oligomerization, pore formation, and membrane translocation. Consistently, we demonstrate a C95A variant of FGF2 to be characterized by a severe secretion phenotype. By contrast, while also important for efficient FGF2 secretion from cells, a second cysteine residue on the molecular surface of FGF2 (C77) is not involved in FGF2 oligomerization. Rather, we find C77 to be part of the interaction interface through which FGF2 binds to the α1 subunit of the Na,K-ATPase, the landing platform for FGF2 at the inner plasma membrane leaflet. Using cross-linking mass spectrometry, atomistic molecular dynamics simulations combined with a machine learning analysis and cryo-electron tomography, we propose a mechanism by which disulfide-bridged FGF2 dimers bind with high avidity to $PI(4,5)P_2$ on membrane surfaces. We further propose a tight coupling between FGF2 secretion and the formation of ternary signaling complexes on cell surfaces, hypothesizing that C95-C95-bridged FGF2 dimers are functioning as the molecular units triggering autocrine and paracrine FGF2 signaling.

## eLife assessment

This manuscript reports **important** findings, demonstrating a critical role for a cysteine-containing dimerization interface in the secretion of FGF2 through an unconventional pathway. The authors

provide **compelling** evidence, combining in vitro biochemical assays with structural simulation. The work will be of interest to researchers working on protein trafficking and secretion.

## Introduction

Beyond the classical ER/Golgi-dependent secretory pathway, multiple alternative mechanisms of protein secretion from cells have been discovered, processes collectively referred to as 'unconventional protein secretion' (UPS) (*Rabouille, 2017*; *Dimou and Nickel, 2018*; *Pallotta and Nickel, 2020*; *Sparn et al., 2022b*). The group of cargo proteins secreted by ER/Golgi-independent pathways is dominated by factors with fundamental functions in physiological processes such as inflammation and angiogenesis, among others, processes that are frequently linked to disease (*Akl et al., 2016*; *Sitia and Rubartelli, 2018*). Prominent examples for cargo proteins making use of unconventional mechanisms of protein secretion are interleukin 1β (IL1β) and fibroblast growth factor 2 (FGF2) (*Nickel and Seedorf, 2008*; *Nickel and Rabouille, 2009*; *Zhang and Schekman, 2013*; *Rabouille, 2017*; *Sparn et al., 2022b*). With regard to soluble cytoplasmic proteins lacking N-terminal signal peptides, two principal pathways have been identified. One of them is characterized by direct protein translocation across the plasma membrane, a process termed UPS type I with FGF2 (*Sparn et al., 2022b*), HIV-Tat (*Rayne et al., 2010*; *Schatz et al., 2018*), Tau (*Katsinelos et al., 2018*; *Merezhko et al., 2018*; *Katsinelos et al., 2021*), and homeoproteins (*Amblard et al., 2020*; *Joliot and Prochiantz, 2022*) being examples. A second pathway mediating unconventional secretion of soluble cytoplasmic cargo proteins known as UPS type III is based on intracellular vesicle intermediates such as autophagosomes or endocytic compartments (*Malhotra, 2013*; *Zhang and Schekman, 2013*; *Rabouille, 2017*; *Dimou and Nickel, 2018*; *Ye, 2018*; *Liu et al., 2020*). Interestingly, certain cargo proteins such as IL1β can be secreted via both type I and type III UPS pathways, depending on cell types and the physiological conditions that apply (*Dupont et al., 2011*; *Liu et al., 2016*; *Evavold et al., 2018*; *Heilig et al., 2018*; *Monteleone et al., 2018*; *Chiritoiu et al., 2019*; *Liu et al., 2020*; *Pallotta and Nickel, 2020*; *Zhang et al., 2020*). These examples reveal the complexity of secretory processes in mammalian cells as being much more diverse than previously assumed.

The unconventional secretory pathway of FGF2 is based on a small number of components, all of which are physically associated with the plasma membrane (*Sparn et al., 2022b*). The machinery can be classified into auxiliary factors and core machinery components. Auxiliary components are the Na,K-ATPase (*Florkiewicz et al., 1998*; *Dahl et al., 2000*; *Ebert et al., 2010*; *Zacherl et al., 2015*) and Tec kinase that binds to PI(3,4,5)P$_3$ at the inner plasma membrane leaflet (*Ebert et al., 2010*; *Steringer et al., 2012*; *La Venuta et al., 2016*), factors that mediate initial steps of this pathway. The Na,K-ATPase has been demonstrated to be the first contact of FGF2 at the plasma membrane mediated by a direct physical interaction between its α1 subunit and FGF2 (*Legrand et al., 2020*). It is believed to serve as a landing platform of FGF2 at the inner plasma membrane leaflet, however, it has also been hypothesized to play an additional regulatory role in coupling FGF2 membrane translocation to the maintenance of the plasma membrane potential (*Lolicato and Nickel, 2022*; *Sparn et al., 2022b*). In addition, Tec kinase has been shown to make direct physical contact with FGF2, resulting in tyrosine phosphorylation of FGF2 at Y81, an interaction that is likely to occur downstream of the Na,K-ATPase (*La Venuta et al., 2015*; *La Venuta et al., 2016*; *Lolicato and Nickel, 2022*; *Sparn et al., 2022b*). This modification has been proposed to regulate the overall efficiency of FGF2 secretion, in particular in the context of cancer development (*Ebert et al., 2010*; *Steringer et al., 2012*; *La Venuta et al., 2015*; *La Venuta et al., 2016*; *Sparn et al., 2022b*).

Once FGF2 is handed over from the Na,K-ATPase and Tec kinase to the phosphoinositide phosphatidylinositol-4,5-bisphosphate (PI(4,5)P$_2$), the core mechanism of FGF2 membrane translocation into the extracellular space is triggered. It was demonstrated that the interaction with PI(4,5)P$_2$ leads to FGF2 oligomerization on membrane surfaces (*Steringer et al., 2012*; *Müller et al., 2015*; *Steringer et al., 2017*), with hexamers being the most prominent oligomeric state linked to membrane pore formation (*Steringer et al., 2017*; *Šachl et al., 2020*). In addition to the high-affinity interaction of PI(4,5)P$_2$ with a positively charged binding pocket containing K127, R128, and K133 (*Temmerman et al., 2008*; *Temmerman and Nickel, 2009*; *Nickel, 2011*; *Steringer et al., 2012*; *Lolicato et al., 2022*), molecular dynamics (MD) simulations suggest FGF2 to locally accumulate about four to five additional PI(4,5)P$_2$ lipids through low-affinity interactions (*Steringer et al., 2017*). Thus,

FGF2 hexamers may accumulate up to 30 PI(4,5)P$_2$ lipids within a highly confined membrane surface area of about 10 nm$^2$. At these sites, given the cone-shaped structure of PI(4,5)P$_2$, local accumulation of PI(4,5)P$_2$ will likely compromise the bilayer architecture and decrease the energy barrier for the formation of toroidal membrane pores (*Lolicato and Nickel, 2022*; *Sparn et al., 2022b*). Indeed, upon PI(4,5)P$_2$-dependent FGF2 oligomerization on membrane surfaces, transbilayer diffusion of membrane lipids has been observed, suggesting a toroidal architecture of the membrane pores that were formed under these conditions (*Steringer et al., 2012*; *Müller et al., 2015*; *Steringer et al., 2017*; *Dimou and Nickel, 2018*; *Pallotta and Nickel, 2020*). With the heparan sulfate chains of glypican-1 (GPC1) containing high-affinity binding sites for FGF2 in close proximity to the membrane surface, FGF2 oligomers have been shown to get captured and disassembled at the outer plasma membrane leaflet, completing FGF2 membrane translocation into the extracellular space (*Zehe et al., 2006*; *Nickel, 2007*; *Sparn et al., 2022a*). This step is unidirectional as heparan sulfate chains compete with high affinity against PI(4,5)P$_2$ for the same binding site on the molecular surface of FGF2 (*Steringer et al., 2017*). Thus, in conclusion, the core machinery of FGF2 membrane translocation consists of PI(4,5)P$_2$ and heparan sulfate chains linked to GPC1 on opposing sites of the plasma membrane. Along with this, FGF2 locally accumulates PI(4,5)P$_2$ molecules through oligomerization concomitant with the formation of a membrane pore with a toroidal architecture (*Pallotta and Nickel, 2020*; *Lolicato and Nickel, 2022*; *Sparn et al., 2022b*).

The goal of the current study was to shed light on the structural principles that govern PI(4,5)P$_2$-dependent oligomerization of FGF2 on membrane surfaces. Particular emphasis was given to the role of intermolecular disulfide bridges known to be present in PI(4,5)P$_2$-triggered FGF2 oligomers. Furthermore, on a functional basis, it was known that a variant form of FGF2 lacking both of the two surface cysteine residues C77 and C95 is incapable of both oligomerization, membrane pore formation, and secretion from cells (*Müller et al., 2015*; *Steringer et al., 2017*; *Dimou and Nickel, 2018*; *Steringer and Nickel, 2018*). However, the way these cysteines contribute to disulfide bridge formation and their specific roles in a cellular context remained unknown. In the current study, through a combination of biochemical, cell biological, and structural techniques including cryo-electron tomography (cryo-ET), cross-linking mass spectrometry (XL-MS), atom-scale biomolecular simulations, and deep learning, we mechanistically dissected the functional roles of C77 and C95 in the sequence of events that constitute the unconventional secretory pathway of FGF2. As opposed to previous concepts, we found C77 not to be involved in FGF2 oligomerization. Rather, along with K54 and K60 (*Legrand et al., 2020*), we revealed a role for C77 in building the molecular interface through which FGF2 binds to the α1 subunit of the Na,K-ATPase, the starting point of this unusual pathway of protein secretion. By contrast, through the formation of disulfide bridges, we found C95 to be essential for PI(4,5)P$_2$-dependent FGF2 oligomerization on membrane surface, producing the dimeric building blocks for higher FGF2 oligomers that can trigger the formation of lipidic membrane pores. Our findings further imply that the heparan sulfate chains of GPC1 containing high-affinity binding sites for FGF2 reverse this process by disassembling membrane pore-forming FGF2 oligomers into C95-C95 disulfide-bridged dimers. The latter are likely to represent the primary ligands for the formation of FGF2 signaling complexes for autocrine and paracrine FGF signal transmission into cells (*Decker et al., 2016*; *Nawrocka et al., 2020*). We, therefore, propose unconventional secretion of FGF2 from cells and autocrine signal transmission into cells to represent tightly coupled processes.

## Results

### Cysteine residues on the molecular surface of FGF2 are required for efficient secretion of FGF2

To elucidate the precise mechanisms that turn C77 and C95 into critical *cis* elements required for unconventional secretion of FGF2 in a cellular context, we analyzed their individual roles in both FGF2 recruitment at the inner plasma membrane leaflet and FGF2 membrane translocation into the extracellular space (*Figure 1*). Using a single molecule TIRF assay established previously (*Dimou et al., 2019*; *Legrand et al., 2020*; *Lolicato et al., 2022*), we found FGF2 mutants lacking either C77, C95, or both C77 and C95 to be impaired in membrane recruitment at the inner leaflet of the plasma membrane (*Figure 1A and B*). This phenomenon was found to be statistically significant for C95A and C77/95A mutants of FGF2, indicating that oligomerization is required for robust FGF2 membrane

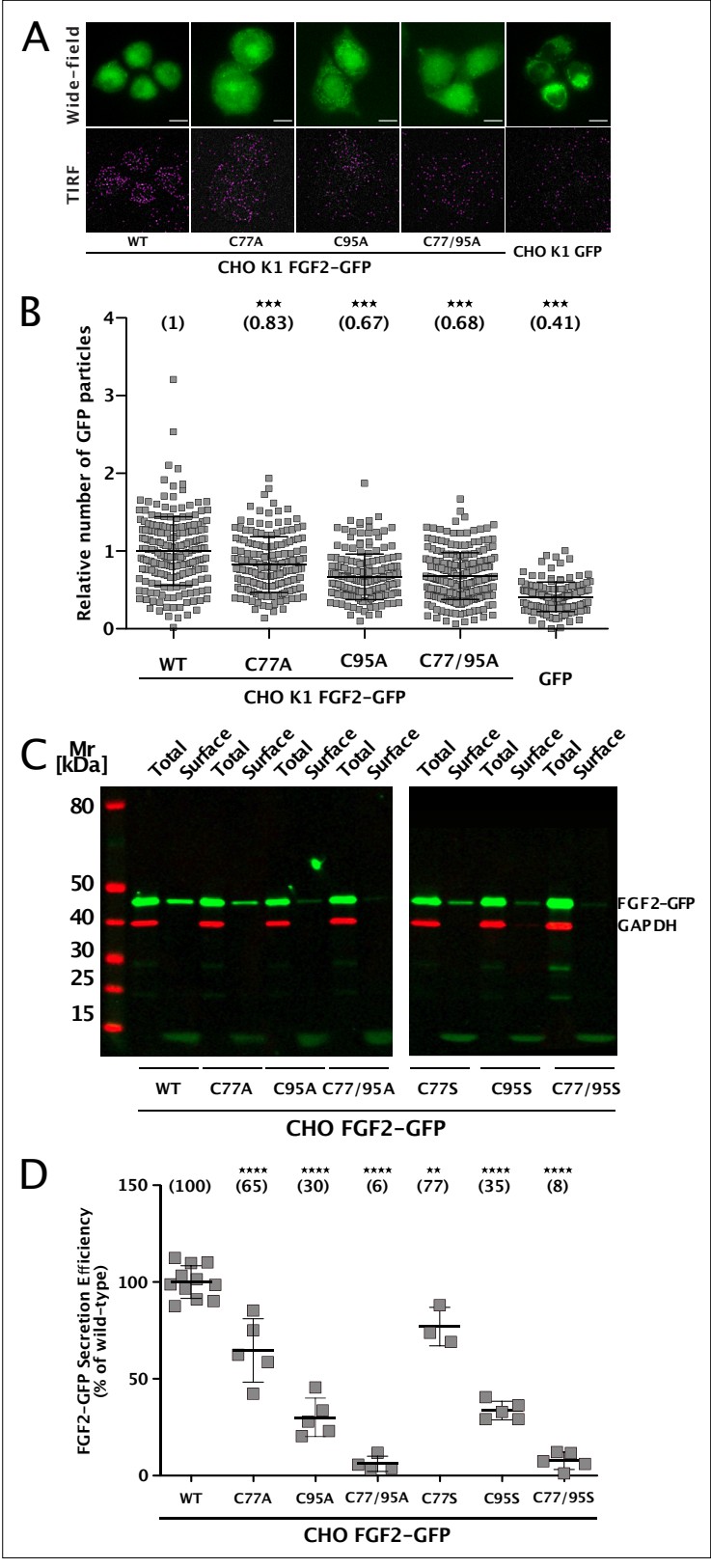

**Figure 1.** Cysteine residues in positions 77 and 95 of fibroblast growth factor 2 (FGF2) play a role in its unconventional secretion from cells. (**A**) Representative wide-field and TIRF images of real-time single molecule TIRF recruitment assay conducted on stable CHO K1 cell lines overexpressing either wild-type (WT) or mutant (C77A, C95A, C77/95A) FGF2-GFP in a doxycycline-dependent manner. Beyond cell lines expressing various forms

*Figure 1 continued on next page*

*Figure 1 continued*

of FGF2-GFP, a GFP-expressing cell line was used to subtract GFP background. Wide-field images show the overall FGF2-GFP (or GFP) expression levels. Single FGF2-GFP (or GFP) particles recruited at the inner plasma membrane leaflet and detected within the TIRF field are highlighted with a pink circle. (**B**) Quantification of real-time single molecule TIRF recruitment assay conducted on the cell lines shown in panel A. Recruitment efficiency at the inner plasma membrane leaflet of FGF2-GFP WT was set to 1. Each square represents a single cell. Mean recruitment efficiency values are shown in brackets. Data are shown as mean with standard deviations (n=4). Statistical analysis was based on a one-way ANOVA test performed in Prism (version 9.4.1), ***$p \leq 0.001$. (**C**) Representative western blot of cell surface biotinylation assay conducted on stable CHO cell lines overexpressing either WT or mutant (C77A, C95A, C77/95A, C77S, C95S, C77/95S) FGF2-GFP in a doxycycline-dependent manner. Total cellular proteins and biotinylated surface proteins were analyzed. The analysis was conducted against GFP, to detect the various FGF2-GFP mutant forms, and GAPDH, both as a loading and a cellular integrity control. (**D**) Quantification of cell surface biotinylation assay conducted on the cell lines shown in panel D. Secretion efficiency of FGF2-GFP WT was set to 100%. Mean secretion efficiency values for each cell line are shown in brackets. Data are shown as mean with standard deviations (n=4). Statistical analysis was based on a one-way ANOVA test performed in Prism (version 9.4.1), not significant (ns) $p > 0.05$, ***$p \leq 0.001$.

The online version of this article includes the following source data for figure 1:

**Source data 1.** Real-time single molecule TIRF recruitment assay.

**Source data 2.** Original file for the western blot analysis in *Figure 1C* (cell surface biotinylation assay).

**Source data 3.** PDF containing *Figure 1C* and original scans of the relevant western blot analysis.

recruitment at the inner plasma membrane leaflet. However, a moderate but consistently observed reduction of membrane recruitment was also observed for C77A. A potential reason for this subtle phenotype might be that this substitution causes a small disturbance of the interface between FGF2 and the α1 subunit of the Na,K-ATPase. To quantitatively assess FGF2 translocation to cell surfaces with the same set of FGF2 variants, we used a well-established cell surface biotinylation assay (*Seelenmeyer et al., 2005*; *Zehe et al., 2006*; *Müller et al., 2015*). Additionally, we also tested FGF2 variants in which cysteines were substituted with serine to exclude the observed phenotypes to depend on alanine substitutions of cysteines. As shown in *Figure 1C, D* C77A or C77S substitution had a mild phenotype that was found to be statistically significant. By contrast, the substitution of C95 to alanine or serine in FGF2 caused a severe secretion phenotype (*Figure 1C and D*). Of note, a combination of C77A (or C77S) and C95A (or C95S) further limited FGF2 secretion to background levels in a highly significant manner, suggesting that both C77 and C95 play important but most likely different roles in unconventional secretion of FGF2 from cells.

## C95 is essential for PI(4,5)P$_2$-dependent formation of FGF2 oligomers

To study the potential roles of C77 and C95 in PI(4,5)P$_2$-dependent FGF2 oligomerization, we conducted both in vitro reconstitution experiments with purified components (*Figure 2*) and FGF2 cross-linking experiments in cells (*Figure 3*). For the first approach, we used giant unilamellar vesicles (GUVs) and purified variant forms of FGF2 as GFP fusion proteins (*Figure 2B*) to quantify FGF2 oligomerization states by fluorescence correlation spectroscopy (FCS)/brightness analyses (*Figure 2A*; *Steringer et al., 2017*; *Šachl et al., 2020*; for details see Materials and methods). To allow for a systematic comparison with previous studies (*Steringer et al., 2012*; *Müller et al., 2015*; *Steringer et al., 2017*), phosphomimetic versions of FGF2 (Y81pCMF) were used under all experimental conditions. Consistent with earlier findings (*Steringer et al., 2017*), the wild-type (WT) version of FGF2-GFP was characterized by an average oligomeric state of about 6–8 subunits (*Figure 2A*). By contrast, an FGF2 variant form lacking both C77 and C95 failed to oligomerize, an observation that again was consistent with previous findings (*Müller et al., 2015*; *Steringer et al., 2017*). Intriguingly, in the continued presence of C77, substituting C95 by alanine severely impaired PI(4,5)P$_2$-dependent FGF2 oligomerization, an effect that was similar to what was observed with an FGF2 C77/C95A double substitution. By contrast, replacing C77 with alanine in the continued presence of C95 did not impact PI(4,5)P$_2$-dependent FGF2 oligomerization to a significant extent (*Figure 2A*).

To challenge these findings in a cellular context, we conducted cross-linking experiments in cellular lysates (*Figure 3*). These studies focused on FGF2 dimers that have previously been shown to be abundantly present at the inner plasma membrane leaflet (*Dimou et al., 2019*). To systematically

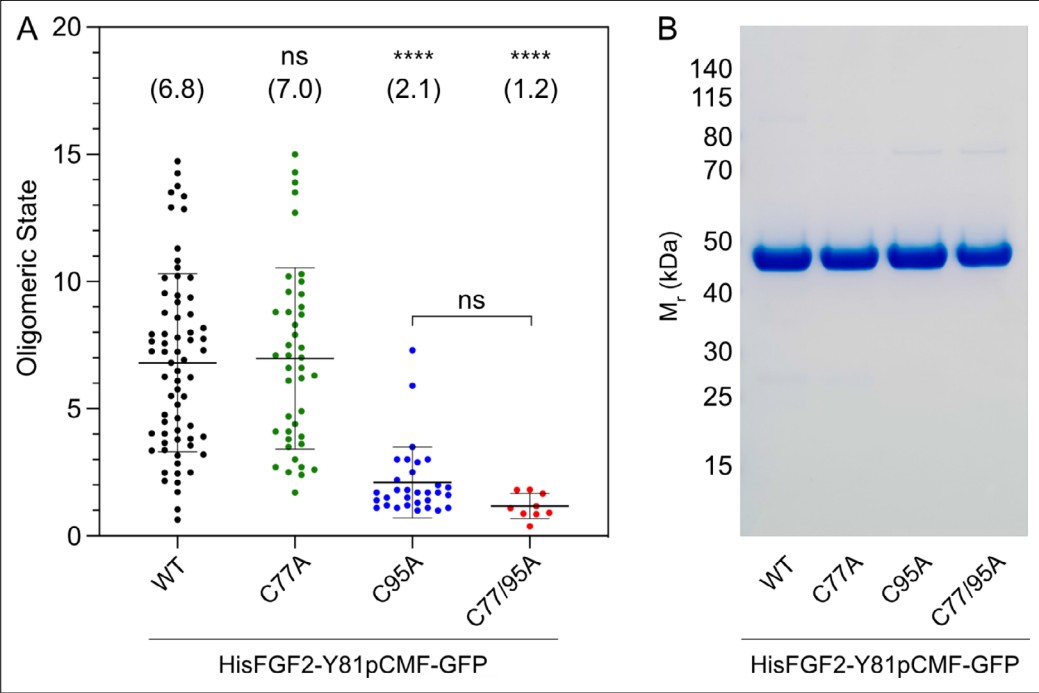

**Figure 2.** Formation of higher fibroblast growth factor 2 (FGF2) oligomers on the membrane surface of giant unilamellar vesicles (GUVs) depends on C95. (**A**) Oligomeric size distribution of FGF2-GFP variants. GUVs with a plasma membrane-like lipid composition containing 2 mol% phosphatidylinositol-4,5-bisphosphate (PI(4,5)P$_2$) were incubated with variant forms of His-tagged FGF2-Y81pCMF-GFP as indicated. The oligomer size was determined by brightness analysis as described in detail in Materials and methods. Each dot corresponds to a data point measured on a single GUV with the number of GUVs analyzed n (wild-type [WT]; n=68, C77A; n=42, C95A; n=31, C77/95A; n=9). Mean values with standard deviations are shown. One-way ANOVA with Tukey's post hoc test was performed in Prism (version 9.4.1). Mean values are shown in brackets, not significant (ns) p>0.05, ****p≤0.0001. Data distribution was assumed to be normal, but this was not formally tested. (**B**) Sodium dodecyl-sulfate polyacrylamide gel electrophoresis (SDS-PAGE) analysis of FGF2-Y81pCMF-GFP variant forms indicated. Purified proteins were analyzed for homogeneity using Coomassie staining.

The online version of this article includes the following source data for figure 2:

**Source data 1.** Oligomeric size distribution of fibroblast growth factor 2 (FGF2)-GFP variants.

**Source data 2.** Original file for the blot analysis in *Figure 2B*.

**Source data 3.** PDF containing *Figure 2B* and original scans of the relevant blot analysis.

---

compare the WT form of FGF2 with the variant forms used in in vitro experiments (*Figure 2A and B*), we transiently expressed FGF2 WT, C77A, C95A, and C77/95A in HeLa S3 cells. We employed an FGF2-P2A-GFP construct, leveraging the 'self-cleaving' P2A peptide to yield stoichiometric production of untagged FGF2 and GFP. GFP was utilized to monitor transfection efficiency. We used three chemical cross-linkers: *N-p*-maleimidophenylisocyanate (PMPI), bismaleimidoethane (BMOE), and bismaleimidohexane (BMH), characterized by different spacer lengths and chemical functionalities (details in Materials and methods). The rationale of these experiments was that bifunctional cross-linkers targeting thiols would link FGF2 subunits into dimers only when the functional cysteine residues are present, replacing the disulfide bridge during oligomerization. As shown for representative examples in *Figure 3A, B, and C* as well as quantified in *Figure 3D, E, and F*, the amounts of cross-linked FGF2 dimers in cellular lysates (labeled with ♦♦) were similar between FGF2 WT and FGF2 C77A. By contrast, both FGF2 C95A and FGF2 C77/95A were severely impaired in dimer formation. In particular, using BMOE and BMH, bifunctional cross-linkers with different spacer lengths that target the thiols of cysteine side chains, efficient cross-linking of FGF2 dimers was observed for FGF2 WT and C77A whereas a strong reduction of cross-linked dimers was found for FGF2 C95A and C77/C95A. This provides direct evidence for a C95-C95 disulfide bridge formed in FGF2 dimerization at the inner plasma membrane leaflet of cells.

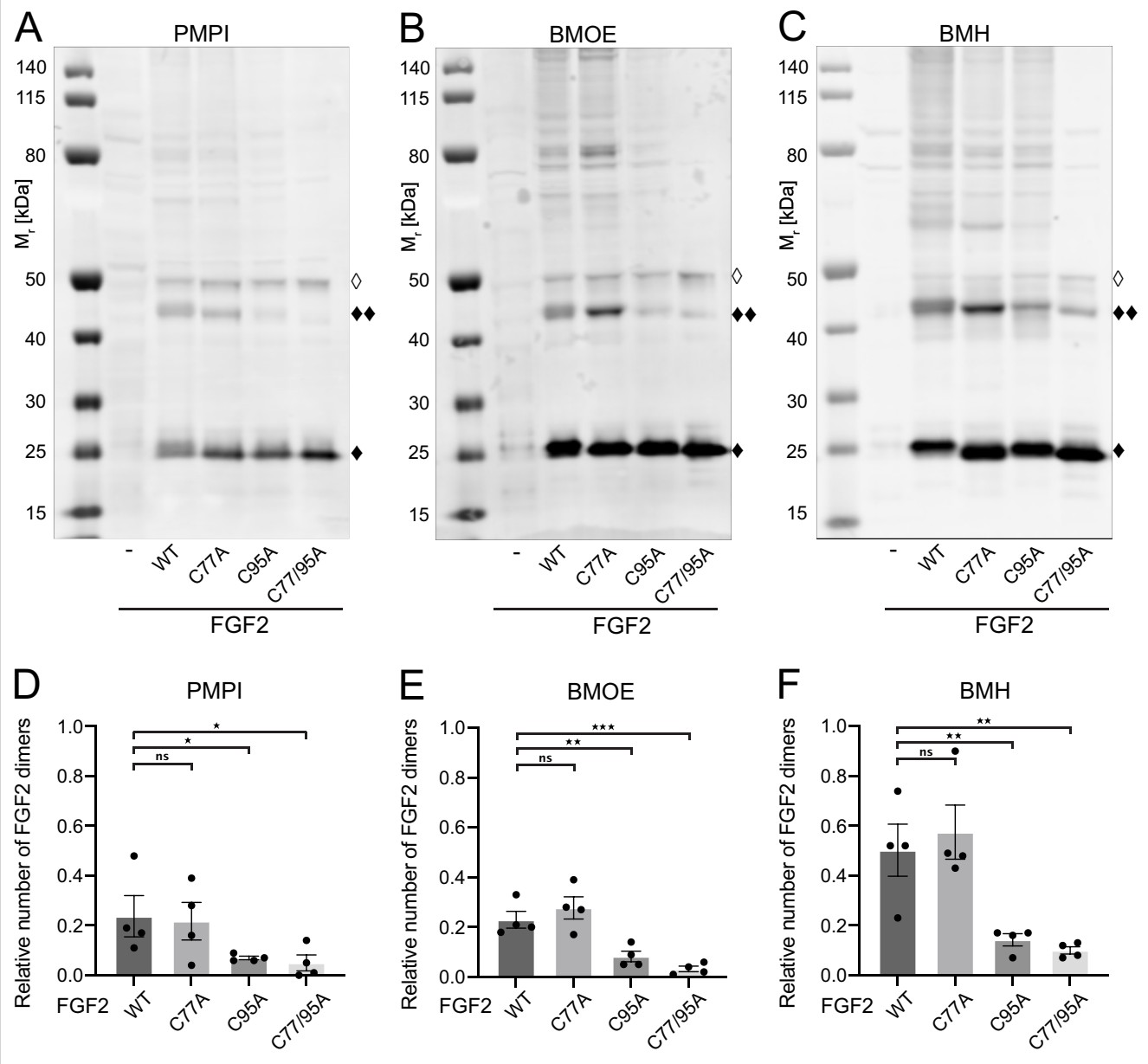

**Figure 3.** Fibroblast growth factor 2 (FGF2) dimer formation in cells depends on C95 as revealed by chemical cross-linking. Using cellular lysates, FGF2 dimer formation was analyzed by chemical cross-linking. The FGF2 variants (wild-type [WT], C77A, C95A, and C77/95A) were transiently expressed in HeLa S3 cells as constructs connecting the FGF2 open reading frame with GFP via a P2A site, producing stoichiometric amounts of untagged FGF2 and GFP, the latter used to label transfected cells. The corresponding cellular lysates were treated with three different cross-linkers: PMPI (*N-p*-maleimidophenylisocyanate; bifunctional cross-linker with a spacer length of 8.7 Å targeting sulfhydryl groups at one end [maleimide] and hydroxyl groups at the other end [isocyanate], A and D), BMOE ([bismaleimidoethane; bifunctional maleimide-based cross-linker with a short 8 Å spacer length targeting sulfhydryl groups], B and E), or BMH ([bismaleimidohexane; bifunctional maleimide-based cross-linker with a long 13 Å spacer length targeting sulfhydryl groups], C and F), respectively. Cross-linking products were analyzed by sodium dodecyl-sulfate polyacrylamide gel electrophoresis (SDS-PAGE) followed by western blotting using polyclonal anti-FGF2 antibodies. (**A, B, C**) Representative examples of the western analyses for each of the three cross-linkers described above. FGF2 monomers (18 kDa) are labeled with '♦', FGF2 dimers (36 kDa) with '♦♦' and small amounts of monomeric full-length FGF2-P2A-GFP (~50 kDa) with '◊'. (**D, E, F**) Quantification of FGF2 dimer to FGF2 monomer ratios. Signal intensities were quantified using a LI-COR Odyssey CLx imaging system. The FGF2 dimer to monomer ratios were determined in four independent experiments with the standard error of the mean shown, not significant (ns) p>0.05, *p≤0.05, **p≤0.01, ***p≤0.001. Statistical analyses were based on a two-tailed, unpaired t-test using GraphPad Prism (version 9.4). Data distribution was assumed to be normal, but this was not formally tested.

The online version of this article includes the following source data for figure 3:

**Source data 1.** Cross-linking quantification of fibroblast growth factor 2 (FGF2) dimer to FGF2 monomer ratios.

*Figure 3 continued on next page*

*Figure 3 continued*

**Source data 2.** Original file for the western blot analysis in *Figure 3A*.

**Source data 3.** PDF containing *Figure 3A* and original scans of the relevant western blot analysis.

**Source data 4.** Original file for the western blot analysis in *Figure 3B*.

**Source data 5.** PDF containing *Figure 3B* and original scans of the relevant western blot analysis.

**Source data 6.** Original file for the western blot analysis in *Figure 3C*.

**Source data 7.** PDF containing *Figure 3C* and original scans of the relevant western blot analysis.

The combined findings from in vitro experiments (*Figure 2*) and cell-based analyses (*Figure 3*) provide compelling evidence for C77 not to be involved in PI(4,5)P$_2$-dependent FGF2 oligomerization. Instead, consistent with previous studies (*Müller et al., 2015*; *Steringer et al., 2017*), these data suggest that membrane-associated FGF2 dimers are formed by homotypic disulfide bridges linking C95 side chains.

## Cysteine 95 in FGF2 is essential for PI(4,5)P$_2$-dependent membrane pore formation

In previous studies, we showed that PI(4,5)P$_2$-dependent FGF2 oligomerization triggers the formation of membrane pores (*Steringer et al., 2012*; *Müller et al., 2015*; *Steringer et al., 2017*). While

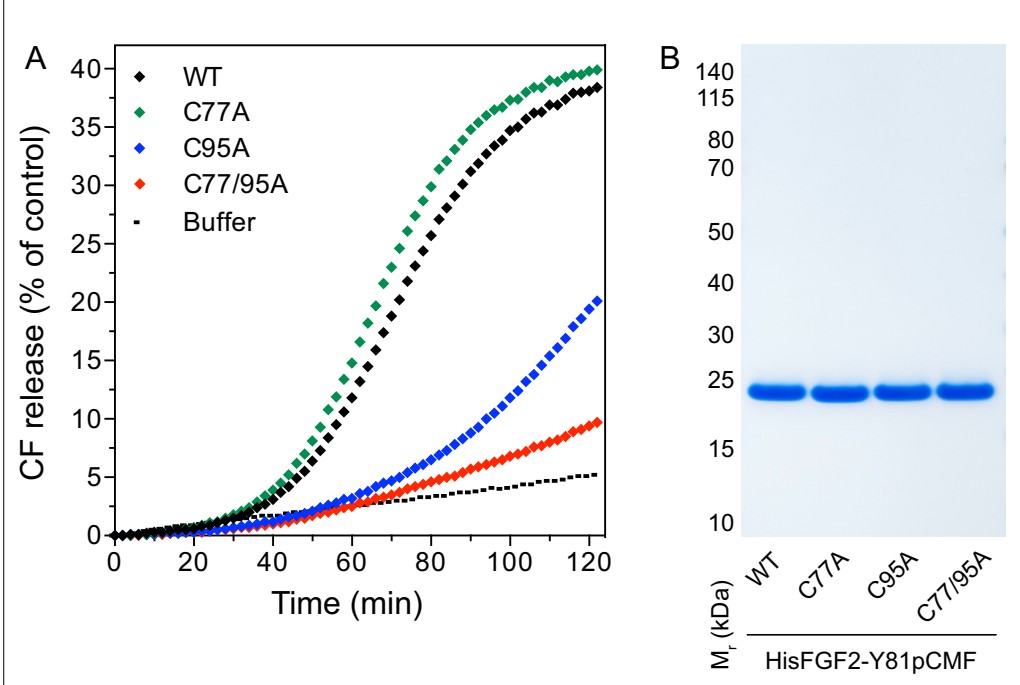

**Figure 4.** Membrane pore formation triggered by fibroblast growth factor 2 (FGF2) oligomers depends on C95. Carboxyfluorescein was sequestered in large unilamellar liposomes containing a plasma membrane-like lipid composition including 2 mol% phosphatidylinositol-4,5-bisphosphate (PI(4,5)P$_2$). (**A**) Liposomes were incubated with the His-tagged FGF2-Y81pCMF (2 μM)-based variant forms indicated (wild-type [WT], C77A, C95A, and C77/95A). Membrane pore formation was analyzed by measuring the release of luminal carboxyfluorescein quantified by fluorescence dequenching as detailed in Materials and methods. The results shown are representative for three independent experiments. (**B**) Quality of the various recombinant proteins was analyzed by sodium dodecyl-sulfate polyacrylamide gel electrophoresis (SDS-PAGE) and Coomassie staining.

The online version of this article includes the following source data for figure 4:

**Source data 1.** Membrane pore formation triggered by fibroblast growth factor 2 (FGF2) oligomers.

**Source data 2.** Original file for the blot analysis in *Figure 4B*.

**Source data 3.** PDF containing *Figure 4B* and original scans of the relevant blot analysis.

we reported previously that this process involves the formation of intermolecular disulfide bridges with a C77/95A variant of FGF2 being inactive (*Müller et al., 2015*; *Steringer et al., 2017*), the way disulfide bridges are formed based on potential contributions from these two cysteine residues remained unclear. As shown in *Figure 4A*, using large unilamellar vesicles (LUVs) and His-tagged versions of recombinant FGF2 variant forms (*Figure 4B*) along with a well-characterized dequenching assay to monitor membrane integrity (*Steringer et al., 2012*; *Müller et al., 2015*), FGF2 WT and FGF2 C77A displayed similar activities efficiently forming membrane pores through which liposome-enclosed fluorophores escape and dilute into the surroundings generating a fluorescent signal based on dequenching. By contrast, similar to the C77/95A variant form of FGF2, substituting C95 with alanine caused a dramatic drop in the formation of membrane pores (*Figure 4A*). These findings are consistent with the experiments shown in *Figures 1–3*, linking FGF2 secretion from cells to membrane pore formation, a process triggered by PI(4,5)P$_2$-dependent FGF2 oligomerization involving C95-mediated formation of disulfide bridges.

## Cysteine 95 is essential for PI(4,5)P$_2$-dependent FGF2 translocation across membranes

To analyze the role of C95 in a comprehensive manner through all steps of the unconventional mechanism of the FGF2 secretion pathway, we completed our in vitro studies by analyzing FGF2 translocation across the membrane of GUVs. These assays were based on previous work reconstituting the ability of FGF2 to physically traverse lipid bilayers based on an inside-out topology setup with PI(4,5)P$_2$ and heparin (located in the lumen of GUVs and used as a surrogate of cell surface heparan sulfate chains) on opposing sides of GUV membranes (*Steringer et al., 2017*). His-tagged GFP fusion proteins of the various FGF2 forms described above (*Figure 2B*) were tested for membrane recruitment, pore formation, and translocation into the lumen of GUVs. As shown in *Figure 5*, GUVs were imaged in three independent fluorescence channels visualizing (i) FGF2 (GFP fluorescence), (ii) GUV lipid bilayers (Rhodamine-PE [Rhod.-PE] fluorescence), and (iii) an Alexa647 fluorophore used as a tracer to monitor membrane integrity. As shown in *Figure 5A* for representative examples, radial intensity profiles were obtained to quantitatively compare fluorescence in the GUV lumen versus the exterior (*Steringer et al., 2017*). For FGF2 WT (*Figure 5A*, subpanel a) and FGF2 C77A (*Figure 5A*, subpanel b), a substantial increase in the GUV lumen could be observed, indicating FGF2 translocation across the membrane. By contrast, for FGF2 C95A (*Figure 5A*, subpanel c) and FGF2 C77/95A (*Figure 5A*, subpanel d), no difference between lumen and exterior could be observed, indicating a failure of FGF2 translocation. Under the conditions indicated, this experimental setup allowed for a statistical analysis of the formation of membrane pores (gray bars in *Figure 5B*) along with luminal accumulation of FGF2-GFP within GUVs (green bars in *Figure 5B*). All GUVs analyzed in *Figure 5* contained PI(4,5)P$_2$ and, where indicated, luminal heparin. Based on the quantification shown in *Figure 5B* and the representative images presented in *Figure 5C*, the dynamic range of the experimental system was apparent from the comparison between GFP fusion proteins containing either FGF2 WT or FGF2 C77/95A. FGF2 WT caused membrane pore formation and, provided the presence of luminal heparin, translocated into the lumen of GUVs. By contrast, FGF2 C77/95A, while getting recruited to the surface of GUVs in a PI(4,5)P$_2$-dependent manner, showed low activity with regard to both membrane pore formation and membrane translocation (*Figure 5A, B, and C*). Under the same conditions, when PI(4,5)P$_2$ was substituted by a Ni-NTA lipid to mediate artificial membrane recruitment via the His tags of all GFP fusion proteins used in these experiments, both FGF2 WT and FGF2 C77/95A were low at background levels with regard to both membrane pore formation and membrane translocation (*Figure 6A, B, and C*). Based on this set of conditions, we analyzed the same parameters for FGF2 C77A and FGF2 C95A. Consistent with our findings documented in *Figures 2–4*, FGF2 C77A behaved similarly to FGF2 WT, efficiently forming membrane pores and, in the presence of luminal heparin, translocating across the membranes of GUVs containing PI(4,5)P$_2$ (*Figure 5A, B, and C*). FGF2 C95A behaved differently, with significantly reduced activities regarding both membrane pore formation and membrane translocation (*Figure 5A, B, and C*). Like FGF2 WT and FGF2 C77/95A, both FGF2 C95A and C77A showed no activities when recruited via the Ni-NTA lipid on GUVs lacking PI(4,5)P$_2$ (*Figure 6A, B, and C*). These findings are consistent with the datasets documented in *Figures 2–4*, demonstrating the essential role of C95 in PI(4,5)P$_2$-dependent FGF2 oligomerization concomitant with membrane pore formation and translocation across lipid bilayers.

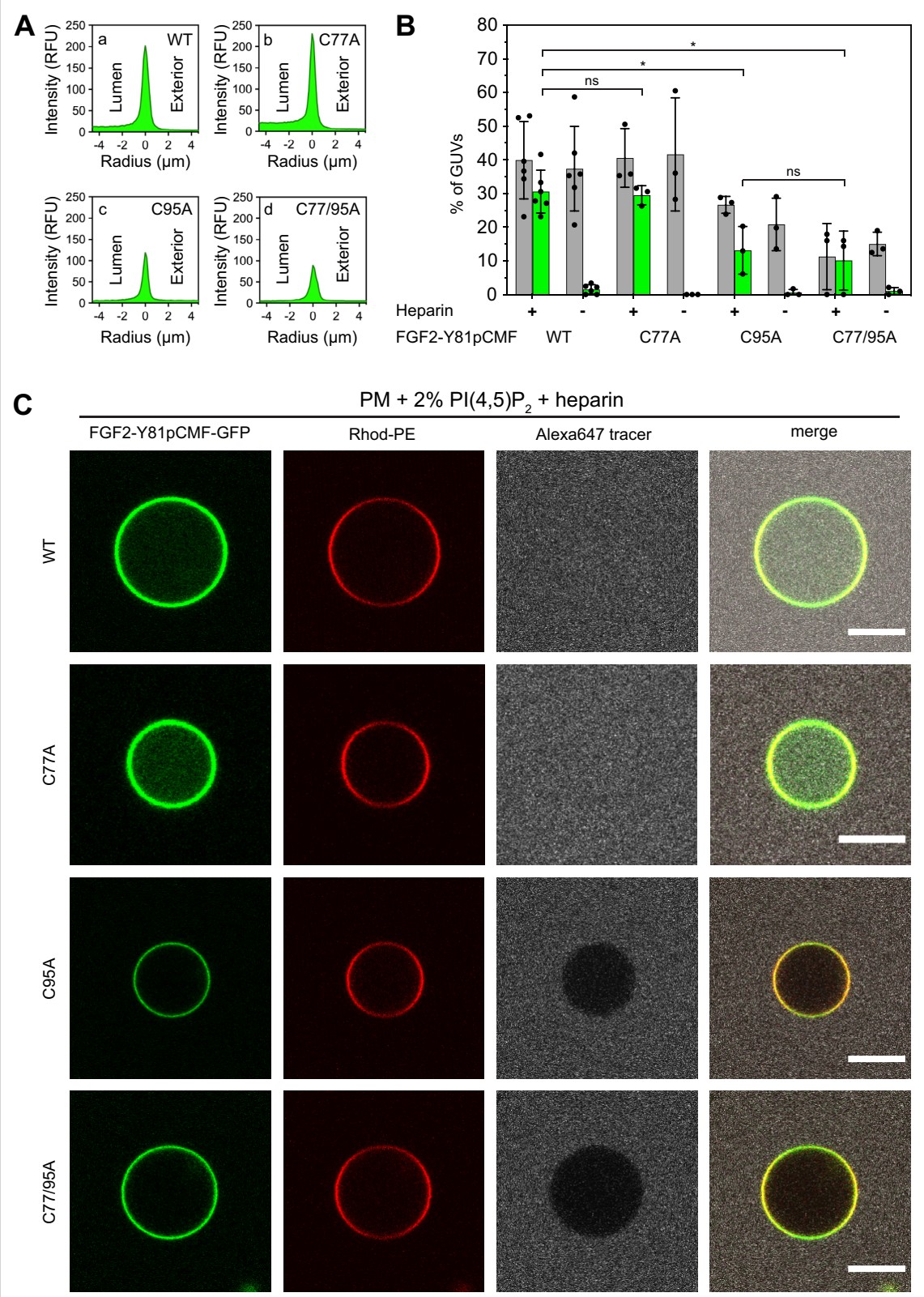

**Figure 5.** Full membrane translocation of fibroblast growth factor 2 (FGF2) across giant unilamellar vesicle (GUV) lipid bilayers depends on C95. Reconstitution of FGF2 membrane translocation with purified components. GUVs with a plasma membrane-like lipid composition containing 2 mol% phosphatidylinositol-4,5-bisphosphate (PI(4,5)P$_2$) were prepared in the presence or absence of long-chain heparins as described in detail in Materials and methods. In brief, Rhodamine-PE (Rhod.-PE) was incorporated into the lipid bilayer during GUV preparation as membrane marker. After removal

*Figure 5 continued on next page*

*Figure 5 continued*

of excess heparin by low-speed centrifugation, GUVs were incubated with His-tagged FGF2-Y81pCMF-GFP (200 nM) variants as indicated and a small fluorescent tracer (Alexa647). Following 180 min of incubation luminal penetration of GUVs by FGF2-Y81pCMF-GFP and small tracer molecules was analyzed by confocal microscopy. (**A**) Radial intensity profiles of representative examples quantifying GFP fluorescence in the GUV lumen versus the exterior for FGF2-Y81pCMF-GFP wild-type (WT) (subpanel a), C77A (subpanel b), C95A (subpanel c), and C77/95A (subpanel d). (**B**) Quantification and statistical analysis of FGF2 membrane translocation and membrane pore formation. Gray bars indicate the percentage of GUVs with membrane pores with a ratio of Alexa647 tracer fluorescence in the lumen versus the exterior of ≥0.6. Green bars indicate the percentage of GUVs where membrane translocation of GFP-tagged proteins had occurred with a ratio of GFP fluorescence in the lumen versus the exterior of ≥1.6 being used as a threshold value. Each dot represents an independent experiment each of which involved the analysis of 20–120 GUVs per experimental condition. Mean values with standard deviations are shown. Statistical analyses are based on two-tailed, unpaired t-test performed in Prism (version 9.4.1), not significant (ns) $p > 0.05$, *$p \leq 0.05$. Data distribution was assumed to be normal, but this was not formally tested. For details, see Materials and methods. (**C**) Representative confocal images of plasma membrane-like GUVs containing $PI(4,5)P_2$ and long-chain heparins in the lumen after 180 min incubation with His-tagged FGF2-Y81pCMF-GFP (200 nM) variants as indicated and a small fluorescent tracer (Alexa647; scale bar = 10 μm).

The online version of this article includes the following source data for figure 5:

**Source data 1.** Fibroblast growth factor 2 (FGF2) membrane translocation and membrane pore formation assay (phosphatidylinositol-4,5-bisphosphate $[PI(4,5)P_2]$) containing liposomes.

## Cysteine 77 is a critical residue at the protein-protein interaction interface between FGF2 and the α1 subunit of the Na,K-ATPase

The experiments shown in *Figures 2–6* demonstrated C77 not to play any role in FGF2 oligomerization and membrane pore formation. Yet, substituting C77 by alanine caused moderate phenotypes in both FGF2 recruitment to the inner plasma membrane leaflet and FGF2 secretion from cells (*Figure 1B and D*). In particular, a C77A substitution significantly enhanced the observed FGF2 secretion phenotype caused by a C95A substitution (*Figure 1D*). When studying the position of C77 on the molecular surface of FGF2, it is evident that this residue is located in spatial proximity to two lysine residues (K54 and K60) that previously have been shown to be part of the protein-protein interface between FGF2 and the α1 subunit of the Na,K-ATPase (*Figure 7A*; *Legrand et al., 2020*). Therefore, in a cellular context, we hypothesized that a substitution of C77 by alanine may interfere with the initial recruitment step of FGF2 at the inner plasma membrane leaflet mediated by the α1 subunit of the Na,K-ATPase, a process that precedes $PI(4,5)P_2$-dependent FGF2 oligomerization and membrane pore formation (*Legrand et al., 2020*). To test this possibility, as shown in *Figure 7*, we used biolayer interferometry (BLI) to study the binding kinetics of various forms of FGF2 toward the domain in the α1 subunit of the Na,K-ATPase to which FGF2 is known to bind (α1-subCD3; *Legrand et al., 2020*). Both α1-subCD3 and the FGF2 variant forms indicated were expressed and purified to homogeneity as recombinant proteins (*Figure 7D*). As detailed in Materials and methods, α1-subCD3 was biotinylated and hooked up on optical sensors coated with streptavidin. To determine kinetic binding parameters between immobilized α1-subCD3 and FGF2, titration experiments were conducted with FGF2 WT concentrations ranging from 1 μM to 15 nM (*Figure 7B*). These experiments revealed this interaction to be characterized by a $K_D$ of 0.1 μM (±0.01), an association constant $k_{on}$ of $1.46 \times 10^4$ M$^{-1}$ × s$^{-1}$ (±0.1 × 10$^4$), and a $k_{off}$ constant of $1.46 \times 10^{-3}$ s$^{-1}$ (±0.15 × 10$^{-3}$). To compare the FGF2 variant forms indicated (FGF2 WT, C77A, C95A, C77/95A, K54/60E-C77A, and K54/60E) with regard to their binding parameters toward α1-subCD3, all of them were used at a concentration of 1 μM, making use of the full dynamic range of the experimental system. As shown in *Figure 7C* and quantified in *Figure 7E*, FGF2 WT and FGF2 C95A efficiently interacted with α1-subCD3. By contrast, FGF2 C77A was severely impaired in binding to α1-subCD3, a phenomenon shared with FGF2 K54/60E-C77A and K54/60E. These findings are consistent with previous observations demonstrating K54 and K60 to be part of the protein-protein interaction interface between FGF2 and α1-subCD3 (*Legrand et al., 2020*). Interestingly, FGF2 C77/95A was more severely impaired in interactions toward α1-subCD3 than FGF2 C77A, an observation that may indicate FGF2 dimerization via C95 to play a role in efficient interactions between FGF2 and the Na,K-ATPase. This may also explain the finding that FGF2 C95A showed a slight but significant reduction in binding efficiency toward α1-subCD3 when compared with FGF2 WT (*Figure 7C and E*). In conclusion, C77, along with K54 and K60, is part of the molecular surface of FGF2 that makes a direct physical contact with α1-subCD3. This, in turn, is consistent with C77A not being involved in $PI(4,5)P_2$-dependent FGF2 oligomerization and membrane translocation, as demonstrated in the in vitro experiments shown in *Figures 2–6*. Rather, cellular phenotypes observed for

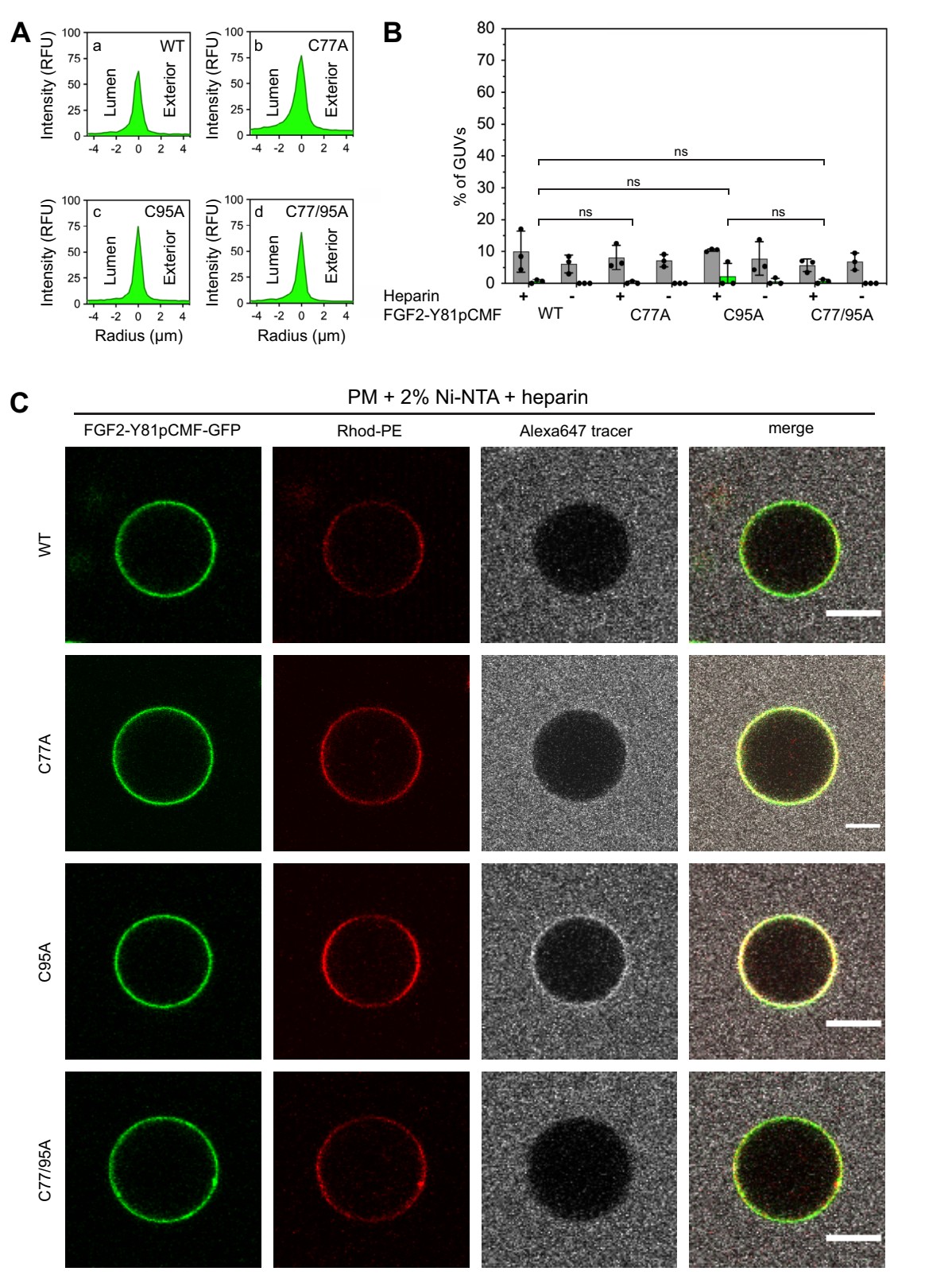

**Figure 6.** Fibroblast growth factor 2 (FGF2) membrane translocation across giant unilamellar vesicle (GUV) lipid bilayers is abrogated when phosphatidylinositol-4,5-bisphosphate (PI(4,5)P$_2$) is substituted by a Ni-NTA lipid used to recruit His-tagged FGF2 fusion proteins. GUVs with a plasma membrane-like lipid composition containing 2 mol% Ni-NTA-lipid anchor were prepared in the presence or absence of long-chain heparins. Luminal penetration of GUVs by FGF2-Y81pCMF-GFP was analyzed by confocal microscopy as described in the legend to *Figure 5*. (**A**) Radial intensity profiles

*Figure 6 continued on next page*

*Figure 6 continued*

of representative examples quantifying GFP fluorescence in the GUV lumen versus the exterior for FGF2-Y81pCMF-GFP wild-type (WT) (subpanel a), C77A (subpanel b), C95A (subpanel c), and C77/95A (subpanel d). (**B**) Quantification and statistical analysis of FGF2 membrane translocation and membrane pore formation. Gray bars indicate the percentage of GUVs with membrane pores with a ratio of Alexa647 tracer fluorescence in the lumen versus the exterior of ≥0.6. Green bars indicate the percentage of GUVs where membrane translocation of GFP-tagged proteins had occurred with a ratio of GFP fluorescence in the lumen versus the exterior of ≥1.6 being used as a threshold value. Each dot represents an independent experiment each of which involved the analysis of 20–120 GUVs per experimental condition. Mean values with standard deviations are shown. Statistical analyses are based on two-tailed, unpaired t-test performed in Prism (version 9.4.1), not significant (ns) p>0.05. Data distribution was assumed to be normal, but this was not formally tested. (**C**) Representative confocal images of plasma membrane-like GUVs containing Ni-NTA-lipid anchor and long-chain heparins in the lumen after 180 min incubation with His-tagged FGF2-Y81pCMF-GFP (200 nM) variants as indicated and a small fluorescent tracer (Alexa647; scale bar = 10 μm).

The online version of this article includes the following source data for figure 6:

**Source data 1.** Fibroblast growth factor 2 (FGF2) membrane translocation and membrane pore formation assay (Ni-NTA containing liposomes).

FGF2 C77A concerning recruitment at the inner plasma membrane leaflet and translocation into the extracellular space (*Figure 1*) can be attributed to impaired binding efficiencies of FGF2 C77A toward the Na,K-ATPase (*Figure 7*), the landing platform that mediates the initial contact of FGF2 with the plasma membrane as the starting point of its transport route into the extracellular space.

## Simulations reveal that the C95-C95 interaction interface dominates the observed dimerization interfaces

The results from the biochemical and cell-based experiments shown in *Figures 1–7* provided direct evidence for a crucial role of C95 in FGF2 dimerization on membrane surfaces based on the formation of disulfide bridges. To investigate the likelihood of this interaction occurring independently of disulfide bond formation, we created 360 initial structures (see 360° Analysis: sampling of the dimerization interface through atomistic MD simulations, Materials and methods) in which two FGF2 monomers (not disulfide-bridged linked) attached to the membrane surface in close proximity underwent different orientations with respect to each other, degree by degree, and each of these systems was simulated for 0.5 μs. The goal was to find out all dimerization interfaces where C95 is involved. Visualization of the simulations immediately revealed that the C95 residues of the monomers sought proximity to each other. This was not observed for C77 residues (*Figure 8A*). Analysis of the simulation data using a combination of dimensionality reduction (*Figure 8B*) and clustering with machine learning techniques revealed that FGF2 dimers formed eight noteworthy clusters (*Figure 8C*). In the cluster with the largest population (Cluster 2, *Figure 8D*), the C95-C95 residues of the two monomers were less than 1 nm apart (*Figure 8E*). This spatial arrangement is stabilized by a network of salt bridges between the two monomers, in which the residues K85, E86, K118, E66, and D98 played a crucial role (*Figure 8F*). Among the observed clusters, this cluster is the only one where the C95-C95 pair is compatible to disulfide bridge formation. The findings reveal that the FGF2 dimer structure depicted in Cluster 2 (*Figure 8F*) forms spontaneously, with the C95-C95 residues being in close proximity and oriented at a specific distance conducive to disulfide bridge formation. This spontaneous configuration does not rely solely on forming the disulfide bridge, suggesting that the covalent bond formation plays a pivotal role in stabilizing the interface during the membrane translocation process.

## Characterization of C95-C95 disulfide-bridged FGF2 dimers employing computational approaches

We continued the atomistic simulations further, aiming to obtain structural insights into this process. We utilized computational approaches to study in detail the protein-protein interface of C95-C95-bridged FGF2 dimers. We generated seven C95-C95 FGF2 dimerization interfaces using three different techniques. As explained in the 'Materials and methods' section in detail, these approaches included atomistic MD simulations performed previously (*Steringer et al., 2017*), the ROSETTA protein-protein docking protocol (*Gray et al., 2003*; *Wang et al., 2005*; *Wang et al., 2007*; *Chaudhury and Gray, 2008*), as well as predictions generated with the AlphaFold2-Multimer v3 package (*Evans et al., 2021*; *Jumper et al., 2021*). To stabilize the dimerization interface found in the initial structures, we conducted 1-μs-long MD simulations in water with a C95-C95 disulfide-linked FGF2 dimer as the starting point (*Figure 9A*, subpanels a and b). These simulations produced a consistent

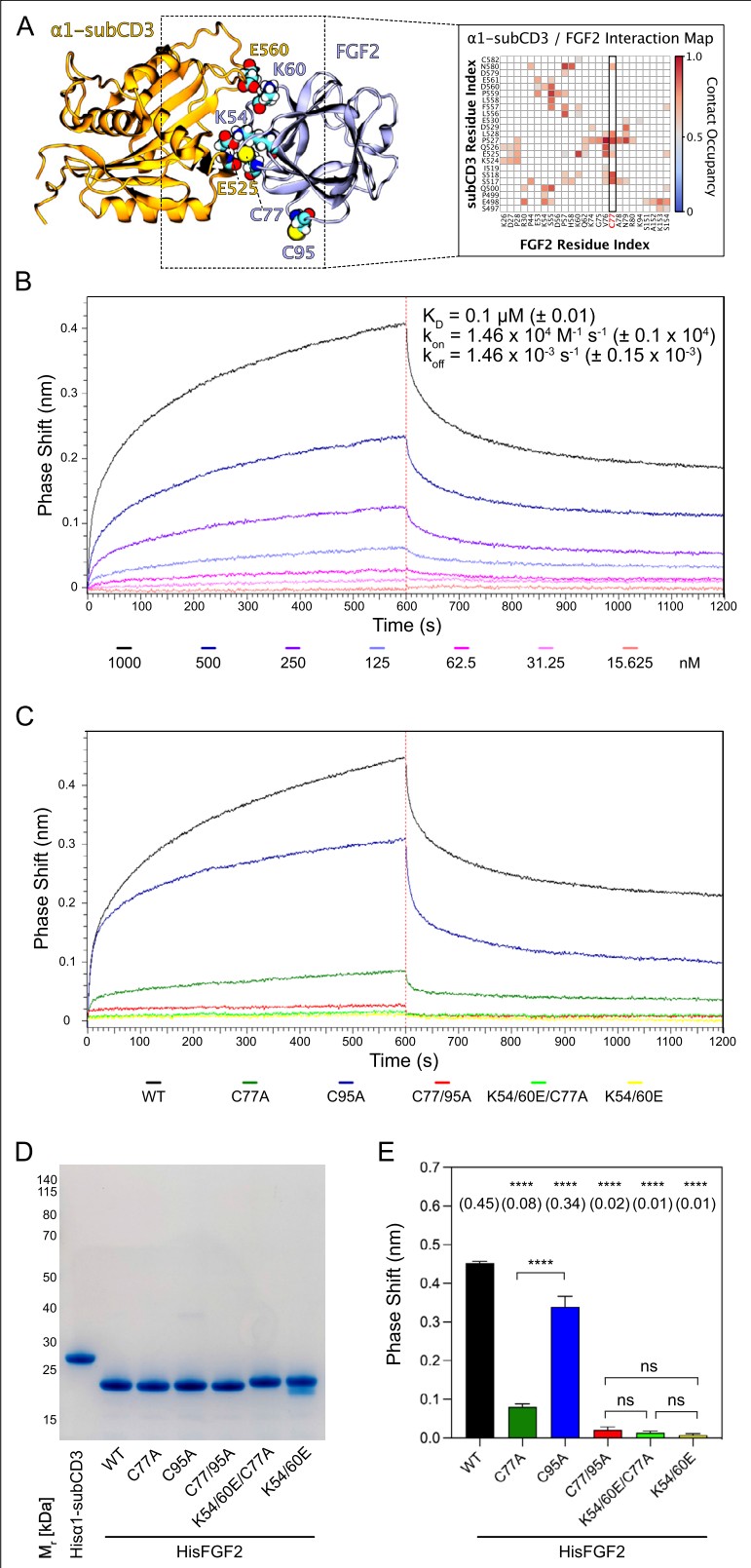

**Figure 7.** C77 is a component of the protein-protein interaction surface between fibroblast growth factor 2 (FGF2) and the α1 subunit of the Na,K-ATPase. Kinetic analysis of the direct interaction of FGF2 with α1-subCD3 (*Legrand et al., 2020*). (**A**) FGF2 binds to α1 with K54, K60, and C77 being part of the protein-protein interaction interface (figure adapted from *Legrand et al., 2020*). (**B**) FGF2 directly binds α1 in a dose-dependent manner. Biolayer

*Figure 7 continued on next page*

*Figure 7 continued*

interferometry (BLI) allows temporal resolution of association and dissociation. Biotinylated Hisα1-subCD3-WT protein was immobilized on Streptavidin sensors followed by incubation with His-tagged FGF2 wild-type protein (HisFGF2-WT) at concentrations indicated. The data shown is representative of three independent experiments. Data were analyzed with Data Analysis HT 12.0 software (Sartorius) using a 1:1 binding model. See Materials and methods for details. Mean with standard deviations of $K_D$, $k_a$, and $k_d$ values (n=3) are given. (**C, D, E**) Comparison of FGF2 variants. (**C**) BLI measurements were conducted using immobilized Hisα1-subCD3 with FGF2 variants (1000 nM concentration) as indicated. The data shown is representative for four independent experiments. (**D**) The quality of HisFGF2 proteins was analyzed by sodium dodecyl-sulfate polyacrylamide gel electrophoresis (SDS-PAGE) and Coomassie staining. 3 µg of each variant were loaded as indicated. (**E**) Phase shift at time point 600 s. Mean values with standard deviations of four independent experiments are shown. One-way ANOVA with Tukey's post hoc test was performed in Prism (version 9.4.1). Mean values are shown in brackets, not significant (ns) p>0.5, ****p≤0.0001. Data distribution was assumed to be normal, but this was not formally tested.

The online version of this article includes the following source data for figure 7:

**Source data 1.** Kinetic analysis of fibroblast growth factor 2 (FGF2)/α1-subCD3 interaction.

**Source data 2.** Original file for the blot analysis in *Figure 7D*.

**Source data 3.** PDF containing *Figure 7D* and original scans of the relevant blot analysis.

interface in five independent simulations, confirming the presence of two ion pairs (E86-K118 and E99-K85) that had previously been suggested to play a role in FGF2 dimerization (*Steringer et al., 2017*). They further revealed that the interface is highly flexible, allowing the two monomers to rotate until they reach a stable conformation (*Figure 9A*, subpanel b). This flexibility is likely to be critical in the context of FGF2-induced pore formation that requires a substantial remodeling of the lipid bilayer. For example, when FGF2 oligomers become accommodated inside toroidal membrane pores, a high degree of freedom is essential to maintain interactions of FGF2 with PI(4,5)P$_2$ in the presence of high membrane curvature. The dimers' final configuration was randomly placed in 10 different orientations, 2 nm away from a POPC membrane surface containing 2 PI(4,5)P$_2$ molecules (*Figure 9B*, subpanel a). These configurations were then simulated for 1 µs. The interaction between the dimer and the membrane preserved the original dimeric interface and revealed the interaction to occur in two steps (*Figure 9B*, subpanels b–c). First, a single FGF2 molecule binds to a single PI(4,5)P$_2$ molecule (*Figure 9B*, subpanel b), followed by the second FGF2 molecule binding to the other PI(4,5)P$_2$ molecule (*Figure 9B*, subpanel c). The free energy profile (*Figure 9B*, subpanel d) indicates that the final state in which both FGF2 subunits were attached to the membrane surface is energetically strongly favorable with a free energy value of –30 kT. The free energy calculations were done using only two PI(4,5)P$_2$ molecules, however, the dimer still had a free energy value comparable to the system where the monomer was bound to five PI(4,5)P$_2$ molecules (*Lolicato et al., 2022*). These findings suggest that the C95 disulfide dimer with a higher avidity has a stronger affinity for the membrane than the FGF2 monomer, even under the conditions that were used for the MD simulations described above. This indicates that the dimer might be more likely to interact with low-abundance PI(4,5)P$_2$ molecules in a cellular context. Additionally, it suggests that FGF2 dimerization might occur before PI(4,5)P$_2$-dependent FGF2 binding to the membrane, possibly triggered by the interaction of FGF2 with the α1 subunit of the Na,K-ATPase, the initial contact of FGF2 with the inner plasma membrane leaflet.

## Characterization of FGF2 dimer interface employing XL-MS

To further analyze the spatial arrangement of the interface that mediates FGF2 dimerization on membrane surfaces, we employed XL-MS (*Figure 10*). A bifunctional cross-linker targeting amino groups in the side chains of amino acids (disuccinimidyldibutyric urea [DSBU]) was used to cross-link FGF2 dimers on liposomal surfaces. Following enzymatic cleavage using the Lys-C protease (see 'Materials and methods' for details), the resulting peptides were subjected to a mass spectrometric analysis. To focus on intermolecular cross-links in the protein-protein interface of membrane-bound FGF2 dimers, we exclusively considered distances produced from identical residues in the subunits of FGF2 dimers. Experiments were conducted with FGF2 WT and FGF2 C77/95A in the presence and absence of PI(4,5)P$_2$-containing liposomes. In *Figure 10A*, under the experimental conditions indicated, cross-linked peptides are provided by the homotypic pairs of amino acid residues they were derived from. Consistent with previous experiments demonstrating FGF2 oligomerization to depend on membrane

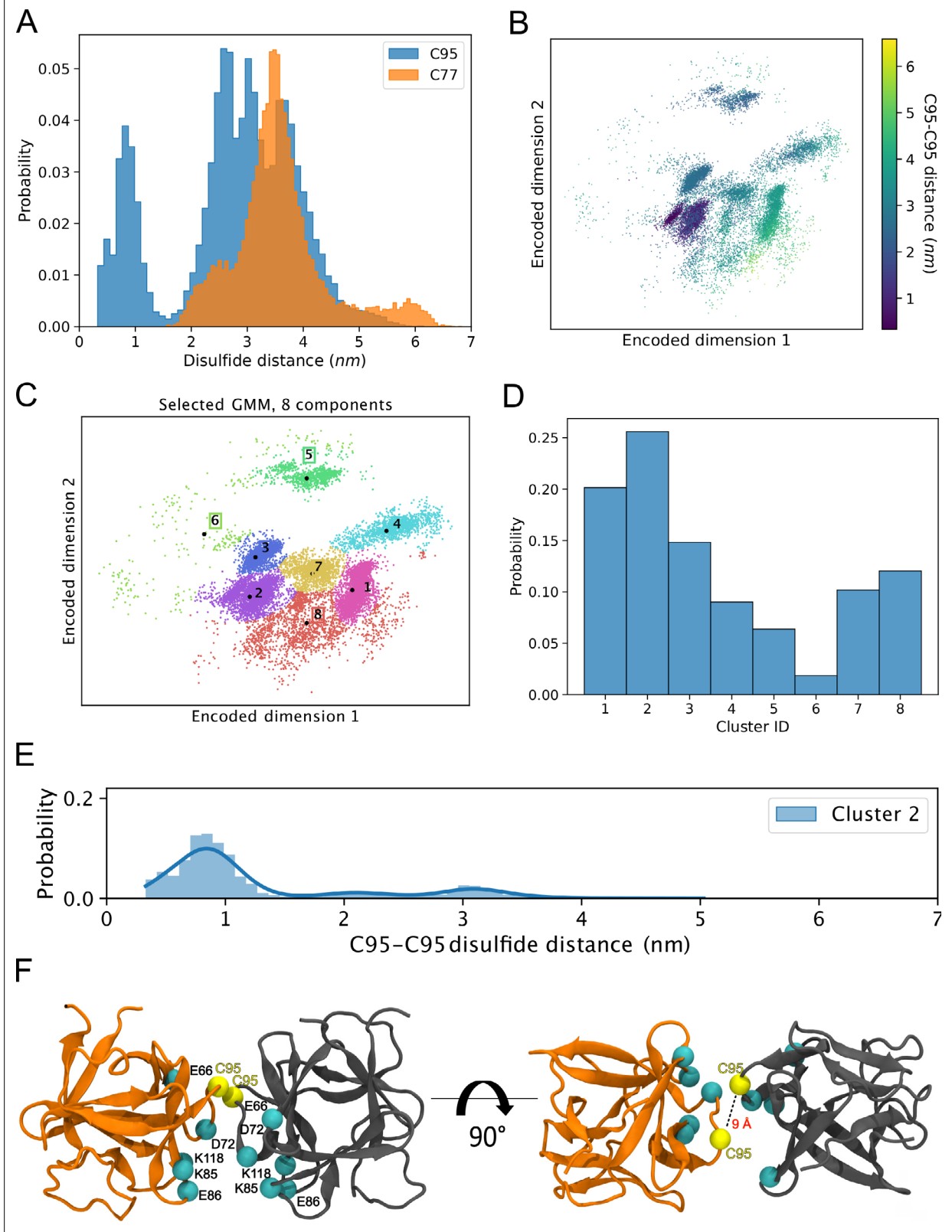

**Figure 8.** Simulations reveal that the C95-C95 interaction interface forms independently of the disulfide bridge. (**A**) The distribution of the C95-C95 and C77-C77 distance shows that only the former can come within 1 nm of each other during the unbiased 360 molecular dynamics (MD) simulations, where one fibroblast growth factor 2 (FGF2) monomer was systematically rotated to explore all possible C95-involved dimerization interfaces. (**B**) The C-alpha atoms of the two monomers were reduced to a two-dimensional (2D) representation using an orthogonal autoencoder. The points are colored with the

*Figure 8 continued on next page*

*Figure 8 continued*

C95-C95 distance. A cluster with low C95-C95 distance is visible in the representation. (**C**) The encoded space was clustered with a Bayesian Gaussian mixture model (GMM) to find regions of distinct conformational structures. The eight identified clusters are indicated in the figure, along with the cluster mean shown in black dots and the corresponding cluster label. (**D**) Populations of the individual clusters in the GMM. Cluster 2 have the largest population among the eight identified clusters. (**E**) The C95-C95 distance indicates that the highest occupied cluster (Cluster 2) also has the highest likelihood of low C95-C95 distance (below 1 nm). (**F**) Representative structure of the FGF2 dimer from Cluster 2 showing C95 residues in proximity and crucial residues responsible for salt bridge interactions.

The online version of this article includes the following source data for figure 8:

**Source data 1.** Populations of the individual clusters in the Bayesian Gaussian mixture model.

surfaces (*Steringer et al., 2012*; *Müller et al., 2015*; *Steringer et al., 2017*), cross-linked peptides were more abundantly found in the presence of PI(4,5)P$_2$-containing liposomes. A structural analysis revealed the collection of cross-linked peptides to be compatible with an FGF2 dimerization interface that brings C95 residues into proximity, enabling the subsequent formation of a disulfide bridge. Specifically, the observed cross-links involving K74-K74, K85-K85, and K94-K94 support an interface that brings C95 residues from two FGF2 molecules into direct contact. These findings are supportive of the MD simulations of the membrane-bound FGF2 dimer as shown in *Figure 10B*, with the Cα-Cα distances of these pairs measuring below the 26.4 Å that are given as DSBU cross-linking distance in rigid molecules (*Iacobucci et al., 2019*; *Piersimoni and Sinz, 2020*). Intriguingly, cross-linked peptides could also be observed with the C77/95A variant form of FGF2, suggesting a protein-protein interface whose formation does not depend on disulfide formation. However, once C95 residues are brought into proximity in FGF2 WT molecules, disulfide formation can take place and further stabilize the interface. The data from the XL-MS experiments further suggest a second independent dimerization interface that may play a role in the formation of higher FGF2 oligomers in which C95-C95-bridged FGF2 dimers are used as building blocks (see average size of FGF2 oligomers to be hexamers as shown in *Figure 2*). This interface is characterized by cross-links involving K34-K34, K54-K54, K60-K60, and K143-K143 and is compatible with a spatial arrangement in which both PI(4,5)P$_2$ binding pockets are pointing to the membrane surface (*Figure 10C*). In conclusion, using an independent experimental approach, the data shown in *Figure 10* directly support the functional experimental data of this study (*Figures 1–7*) and MD simulations (*Figures 8 and 9*), pointing at an FGF2 dimerization interface that brings C95 residues in close proximity that is compatible with the formation of a disulfide bridge.

## Visualization of membrane-associated FGF2 dimers by cryo-ET

To visualize interactions of FGF2 with membrane surfaces at a molecular scale, we conducted cryo-ET using FGF2 bound to liposomes in a PI(4,5)P$_2$-dependent manner. Since FGF2 has a low molecular weight (18 kDa) and thus is challenging to be imaged by cryo-ET, we took advantage of the fact that FGF2 fusion proteins are functional in both cell-based assays (*Figure 1*) and in vitro reconstitution experiments (*Figures 2 and 5*). Since the Halo tag (33 kDa) with its strongly acidic isoelectric point decreases liposome tethering (*Lolicato et al., 2022*), we used a His-FGF2-Y81pCMF-Halo fusion protein with a molecular weight of approximately 51 kDa. FGF2-Y81pCMF-Halo (10 µM) was incubated with LUVs (2 mM lipids with a plasma membrane-like composition containing PI(4,5)P$_2$; *Temmerman et al., 2008*; *Temmerman and Nickel, 2009*; *Steringer et al., 2012*; *Müller et al., 2015*; *Steringer et al., 2017*) for 4 hr at 25°C. Proteoliposomes were vitrified by plunge freezing into liquid ethane (Vitrobot, Thermo Fisher Scientific) as described in 'Materials and methods'. Tilt series were acquired at Krios-GIF-K2 either in-focus using a Volta phase plate (VPP) or at nominal defocus (–3 µm) without the VPP. Cryo-electron tomograms revealed small densities of His-FGF2-Y81pCMF-Halo bound to the membrane surface of LUVs (*Figure 11A*, subpanels a–c). Based on the number of visible Halo domains contained in single particles, our analysis revealed monomers (e.g. *Figure 11A*, subpanel d), dimers (e.g. *Figure 11A*, subpanel e), and higher oligomers (e.g. *Figure 11A*, subpanel f) of His-FGF2-Y81pCMF-Halo. In the latter case, Halo tags could be observed on both sides of the membrane, suggesting that higher His-FGF2-Y81pCMF-Halo oligomers are capable of spanning the lipid bilayer (*Figure 11A*, subpanel f).

Interestingly, dimeric forms of His-FGF2-Y81pCMF-Halo bound to PI(4,5)P$_2$-containing liposomes appeared as V-shaped structures (*Figure 11A*, subpanel e). This finding prompted us to analyze these dimers in more detail, using subtomogram averaging of subvolumes extracted from cryo-electron

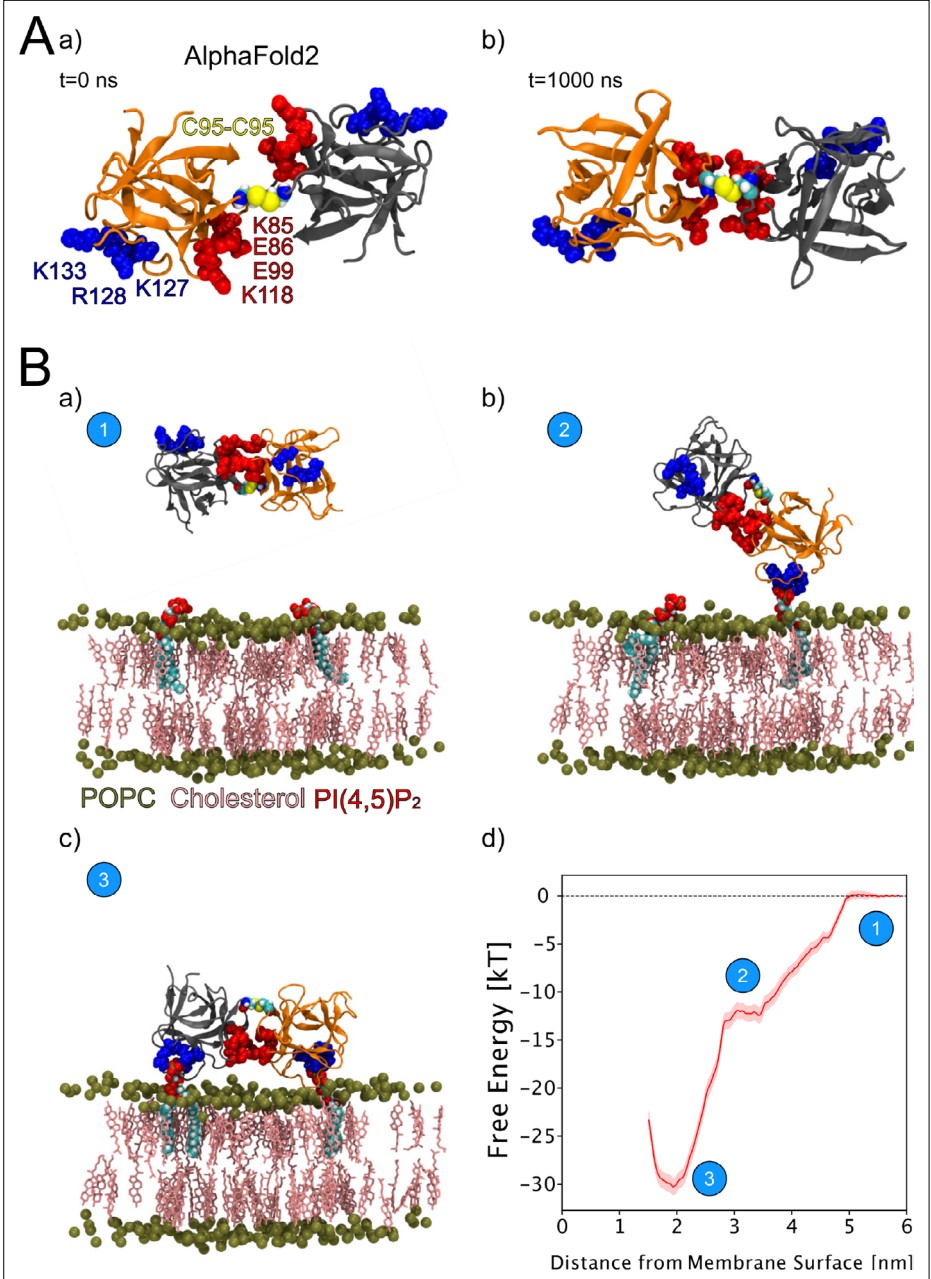

**Figure 9.** Characterization of C95 disulfide-bridged fibroblast growth factor 2 (FGF2) via molecular dynamics (MD) simulations. (**A**) The AlphaFold2 model dimeric interface's stability (subpanel a) was tested by conducting 1-µs-long MD simulations in water, which revealed the interface's high flexibility (subpanel b). (**B**) Unbiased all-atom MD simulations were used to sample the FGF2 dimer-membrane interaction pathway, mediated via the experimentally known phosphatidylinositol-4,5-bisphosphate ($PI(4,5)P_2$) binding pocket (K127, R128, K133). The free energy profile of FGF2 dimer-membrane interaction was determined from biased (umbrella sampling) MD simulations and plotted against the center of the mass distance of FGF2 dimer from phosphate atoms of the interacting membrane surface. Subpanels a–c show the interaction pathway's initial, intermediate, and final states, while subpanel d shows the free energy profile. The statistical error was determined with 200 bootstrap analyses.

The online version of this article includes the following source data for figure 9:

**Source data 1.** Umbrella sampling free energy profile.

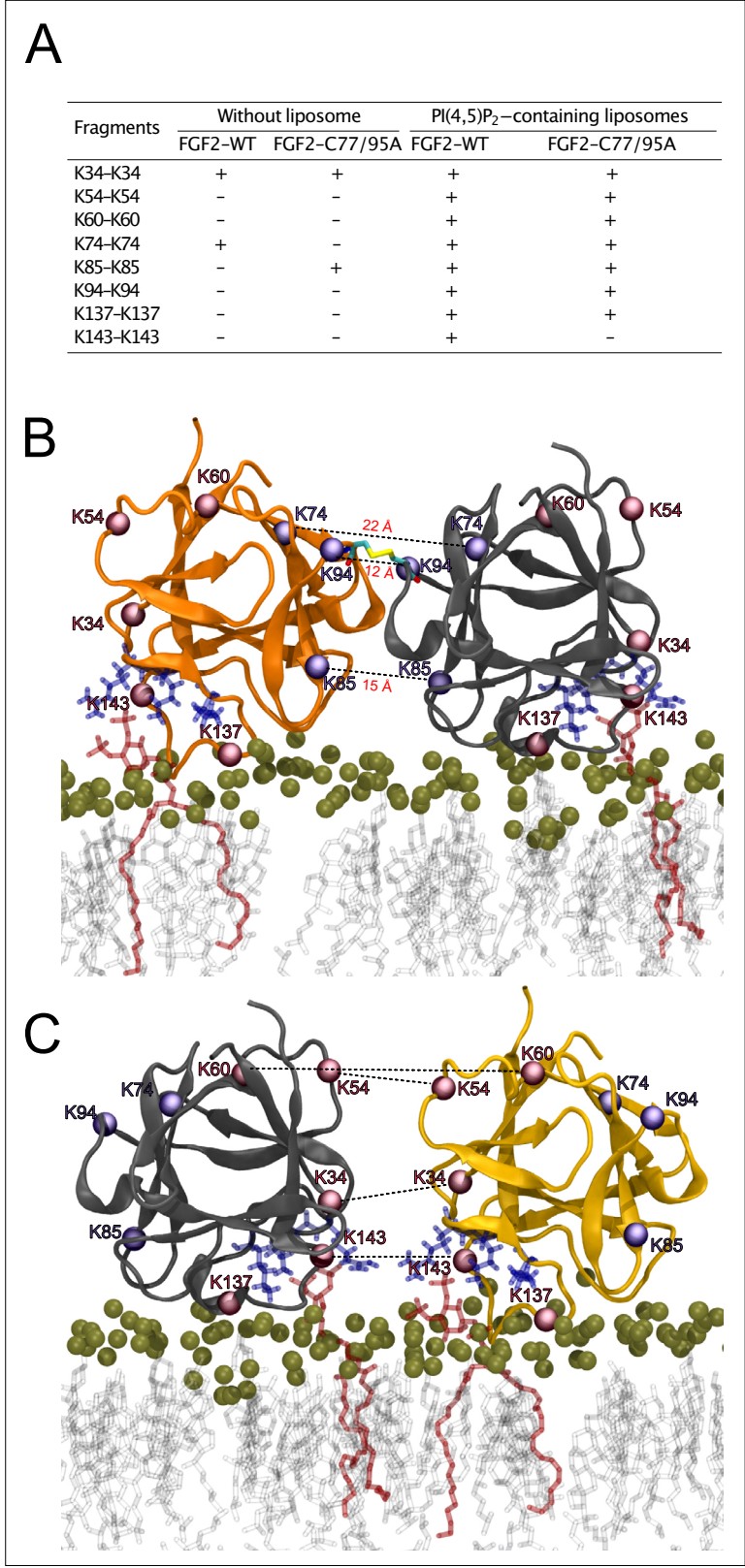

**Figure 10.** Cross-linking mass spectrometry visualization of fibroblast growth factor 2 (FGF2) dimer interfaces. (**A**) Overview of the inter-cross-linked fragments for His-tagged FGF2-WT and FGF2-C77/95A in the presence and absence of phosphatidylinositol-4,5-bisphosphate (PI(4,5)P$_2$)-containing liposomes. (**B**) Liposome-induced cross-linking findings align seamlessly with molecular dynamics simulations' C95-dependent dimer model

*Figure 10 continued on next page*

*Figure 10 continued*

interface. Fragments K74-K74, K85-K85, and K94-K94 can be positioned within the simulation model at a Cα-Cα distance below 23 Å. This concurs with the theoretical maximum Cα-Cα distance of approximately 26.4 Å for disuccinimidyldibutyric urea (DSBU)-linked lysine residues. (**C**) Cross-linking mass spectrometry data are compatible with an additional dimerization interface. It is incompatible with disulfide bridge formation but consistent with a membrane-bound FGF2 dimer configuration. Fragments K34-K34, K54-K54, K60-K60, and K143-K143 can be positioned at a Cα-Cα distance below 23 Å.

The online version of this article includes the following source data for figure 10:

**Source data 1.** Cross-linking mass spectrometry fragments.

tomograms obtained with VPP (*Figure 11B*). A total of 184 V-shaped particles, which showed a similar structure to the one presented in *Figure 11A*, were picked manually using a dipole model and subjected to subtomogram averaging workflow using Dynamo (*Castaño-Díez et al., 2012*; *Castaño-Díez, 2017*; *Castaño-Díez et al., 2017*; *Navarro et al., 2018*; for details, see 'Materials and methods'). The resulting twofold symmetrized subtomogram average was visualized as a three-dimensional (3D) volume (*Figure 11B*; subpanels c–d) in ChimeraX (*Pettersen et al., 2021*). Using the known crystal structures of the Halo domain (PDB:4KAJ) and the FGF2 monomer (PDB:1BFF), we were able to manually fit two FGF2 monomers to the membrane proximal region and two Halo domains protruding away from the membrane surface into the subtomogram average. This indicates that the V-shaped particle corresponds to an FGF2-Y81pCMF-Halo dimer where FGF2 is interacting with the membrane surface of the liposome. However, due to a limited number of particles and heterogeneity in oligomeric states, the resolution of the average is limited and not sufficient to provide direct information about the FGF2 dimerization interface. Therefore, we used AlphaFold2-Multimer (v3) to generate FGF2-HALO dimers that fit into the 3D density in an accurate manner. Remarkably, four of the top-ranked structures resembled a V-shaped FGF2-Halo dimer. However, among these structures, only the fourth-ranked one contained the experimentally known PI(4,5)P$_2$ binding pocket in an orientation that is compatible with PI(4,5)P$_2$-dependent membrane binding. Intriguingly, the C95 residues from the two FGF2 subunits were found in the interaction interface (*Figure 12A and B*). Using this structure as a starting point, we substituted the predicted FGF2 dimer with the one obtained from the MD simulations (*Figures 8 and 9*). In addition, AlphaFold2-Multimer (v3) predicted the positions of the two Halo domains in a way that was not compatible with the electron densities obtained from cryo-ET (*Figure 12A and B*). Therefore, we performed a rigid body fit using ChimeraX software to place the single Halo domains into the density map, using the AlphaFold-Multimer structure as a template. Finally, the modeled V-shaped dimer was positioned on the surface of a model membrane and subjected to a 500 ns simulation. As shown in *Figure 11—video 1*, the dimer was stable over the simulated time period. As depicted in *Figure 11B* (subpanels e–f), the resulting FGF2-Halo dimer is based on a consistent dataset combining cryo-ET data, structural predictions from AlphaFold2-Multimer (v3) and MD simulations. The data provide an initial structural understanding of how FGF2 dimerizes on membrane surfaces. The proposed dimerization interface is compatible with the functional data from both cell-based studies and in vitro experiments presented in this study.

## Discussion

The principal machinery and basic aspects of the molecular mechanism by which FGF2 can physically traverse the plasma membrane to get access to the extracellular space have been revealed in great detail in recent years (*Dimou and Nickel, 2018*; *Pallotta and Nickel, 2020*; *Sparn et al., 2022b*). However, the mechanism by which FGF2 oligomerizes in a PI(4,5)P$_2$-dependent manner on membrane surfaces concomitant with the formation of a lipidic membrane pore continued to be a mystery. In particular, the molecular events remained elusive by which the highly dynamic process of PI(4,5)P$_2$-dependent FGF2 oligomerization triggers the remodeling of the rather stable plasma membrane lipid bilayer into a lipidic membrane pore with a toroidal architecture, the intermediate structure through which FGF2 oligomers can move toward the extracellular space. This process concludes in a GPC1-dependent manner, a cell surface heparan sulfate proteoglycan that captures FGF2 oligomers as they penetrate PI(4,5)P$_2$-dependent membrane pores, followed by disassembly into FGF2 signaling modules on cell surfaces (*Zehe et al., 2006*; *Nickel, 2007*; *Sparn et al., 2022a*). To address the

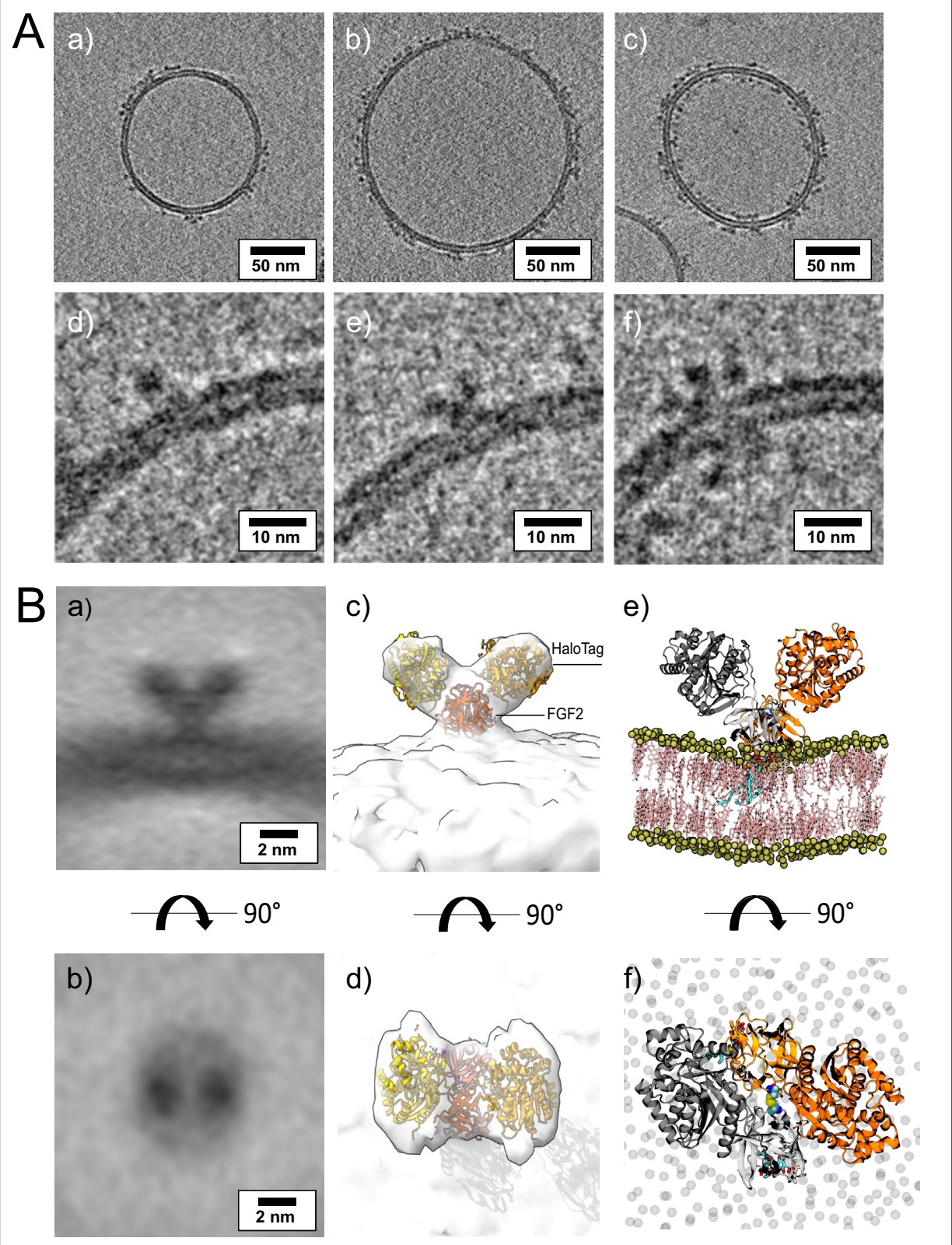

**Figure 11.** Cryo-electron tomography visualization of fibroblast growth factor 2 (FGF2) dimer in proteoliposomes. (**A**) Example slices of cryo-electron tomograms (subpanels a–c; 2.3 nm thickness) showing His-FGF2-Y81pCMF-Halo bound to phosphatidylinositol-4,5-bisphosphate (PI(4,5)P$_2$)-containing liposomes acquired at nominal defocus –3 μm. Magnified views from FGF2-Halo monomers, dimers, and higher oligomers (subpanels e–f). (**B**) Subtomogram average of the V-shaped FGF2-Halo dimer interacting with the membrane of PI(4,5)P$_2$-containing liposomes (subpanels a–b).

*Figure 11 continued on next page*

*Figure 11 continued*

Subtomogram average of 'V-shaped' FGF2 dimers were manually picked using a dipole model in Dynamo (number of particles = 186). Top and side views are shown in subpanels a and b, respectively. (**c–d**) Three-dimensional (3D) map with manually fitted crystal structures of two Halo domains (PDB:4KAJ) and two FGF2 (PDB:1BFF) monomers. (**e–f**) Atom-scale molecular dynamics simulation model of V-shaped C95 disulfide-bridged FGF2-Halo dimer stable over 500 ns.

The online version of this article includes the following video for figure 11:

**Figure 11—video 1.** Simulation of the modeled V-shaped dimer interacting with the membrane surface over 500 ns.

https://elifesciences.org/articles/88579/figures#fig11video1

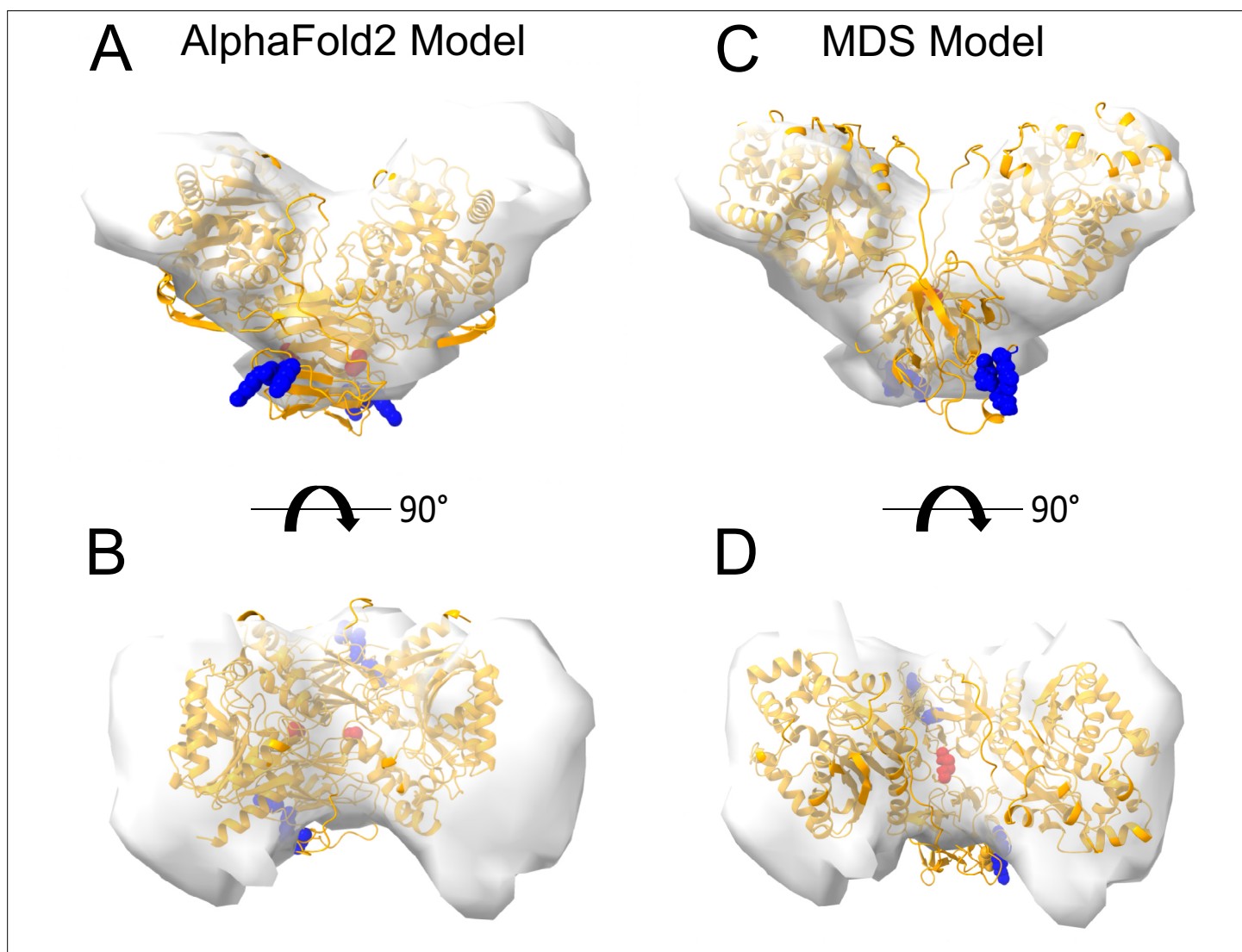

**Figure 12.** Cryo-electron tomography visualization of fibroblast growth factor 2 (FGF2) dimer in proteoliposomes. From a top and side view perspective, a comparison between the AlphaFold2 Multimer v3 model (panels **A–B**) and the one used for molecular dynamics (MD) simulations (panels **C–D**). The AlphaFold model accurately predicted the orientation of FGF2 dimer, with the phosphatidylinositol-4,5-bisphosphate (PI(4,5)P$_2$) binding residues correctly positioned for a membrane-bound state (blue residues). However, the two cysteines 95 (red residues), although located at the interface, were observed to be distant. To improve the model, we replaced the FGF2 dimer with the C95-C95 disulfide-bridged dimer interface characterized with MD simulations (*Figure 9*). Furthermore, we used the 'Fit in Map' command provided by the ChimeraX software to locally optimize the fit of one of the two Halo domains' atomic coordinates into the density map.

long-term goal of a mechanistic understanding of how FGF2 oligomers can trigger the biogenesis of lipidic membrane pores in a PI(4,5)P$_2$-dependent manner, it is crucial to understand how FGF2 molecules assemble into oligomers at the inner leaflet of plasma membranes.

In previous work, we identified two surface cysteines in positions 77 and 95, respectively, that are fully conserved among all mammalian forms of FGF2 (*Müller et al., 2015*; *Steringer et al., 2017*). Of note, these cysteines cannot be found in FGF family members containing N-terminal signal peptides for ER/Golgi-dependent secretion, suggesting a specialized role in unconventional secretion of FGF2 (*Steringer and Nickel, 2018*). Manipulation of these cysteine residues by either NEM-mediated alkylation or substitution by alanines resulted in the inability of FGF2 to oligomerize on membrane surfaces in a PI(4,5)P$_2$-dependent manner (*Müller et al., 2015*). Such conditions further caused a failure of FGF2 to form membrane pores, to translocate across membranes and to get secreted from cells (*Müller et al., 2015*; *Steringer et al., 2017*). When oligomeric species of FGF2 were analyzed by native gel electrophoresis or by comparing migration behavior by reducing and non-reducing sodium dodecyl-sulfate polyacrylamide gel electrophoresis (SDS-PAGE), evidence could be collected that intermolecular disulfide bridges play a role in PI(4,5)P$_2$-dependent FGF2 oligomerization on membrane surfaces (*Müller et al., 2015*). However, it remained unclear as to whether both cysteines are involved and how disulfide bridges are arranged to form FGF2 dimers and higher oligomers.

With the current study, we now obtained key insights into the structural foundations of how FGF2 dimerizes on membrane surfaces. In conjunction with functional in vitro studies and cell-based experiments, this became possible only through a combination of cryo-ET, atomistic MD simulations, machine learning analyses, structural predictions from AlphaFold2-Multimer, and insights from cross-linking mass spectrometry experiments, providing a dataset suited to solve this challenging problem. In particular, we revealed an FGF2 dimerization interface that contains a disulfide bridge that depends on C95, one of two cysteine residues on the molecular surface of FGF2. Of note, disulfide bridge formation driving FGF2 oligomerization was found to be completely independent of C77, a second residue with a thiol side chain that is localized nearby C95 on the molecular surface of FGF2. Intriguingly, our data from machine learning analysis (*Figure 8*), atomistic MD simulations (*Figure 9*), and XL-MS (*Figure 10*) indicate that this dimerization interface assembles independently of C95-C95 disulfide formation, meaning the stability of this configuration is not derived solely from the formation of the disulfide bridge. Instead, it offers kinetic stability essential for disulfide bridge formation at the inner leaflet of the plasma membrane, indicating that the formation of disulfide-bridged FGF2 dimers plays a critical role in stabilizing the dimerization interface during FGF2 membrane translocation. This, in turn, explains why C95 substitutions by alanine or serine cause severe FGF2 secretion phenotypes. These findings clarify the molecular configuration by which disulfide bridges are formed in membrane-associated FGF2 dimers. Furthermore, the inability of C77 to form disulfide bridges had important implications of the overall structure of higher FGF2 oligomers. In particular, we propose a second non-covalent FGF2-FGF2 interaction interface to exist, resulting in the formation of higher FGF2 oligomers in which disulfide bridges and non-covalent FGF2-FGF2 interaction interfaces occur in an alternating arrangement. The C95-dependent disulfide-mediated interface could be revealed in this study (*Figures 8, 9A and B–11*). The second disulfide-indendent interface is likely to be represented by the one identified in this study through XL-MS experiments (*Figure 10A and C*). The combination of these interfaces will be key for future studies, revealing the structure-function relationship of higher FGF2 oligomers forming lipidic membrane pores during FGF2 translocation into the extracellular space.

The limitation to just one type of disulfide bridge in FGF2 oligomers also has implications on how these oligomers are disassembled on cell surfaces, mediated by GPC1 (*Sparn et al., 2022a*; *Sparn et al., 2022b*). We propose the product of this process to be C95-C95-linked FGF2 dimers, generated by the GPC1-dependent disruption of the predicted second non-covalent FGF2-FGF2 interface. Of note, recombinant forms of C95-C95 FGF2 dimers stabilized by chemical cross-linking have been demonstrated to be efficient FGF2 signaling modules (*Decker et al., 2016*; *Nawrocka et al., 2020*). These observations led us to hypothesize that the unconventional secretory pathway of FGF2 and the generation of FGF2 signaling units are tightly coupled processes, with the dimeric FGF2 signaling modules that drive FGF2 signaling already forming at the inner plasma membrane leaflet. In this concept, FGF2 dimers transiently oligomerize into higher oligomers as intermediates of FGF2 membrane translocation, a process that has been shown to occur within a time interval of just 200

ms (*Dimou et al., 2019*). By the action of GPC1, we propose membrane-inserted FGF2 oligomers to be reconverted into FGF2 dimers at the outer leaflet, the principal units of FGF2 signal transduction (*Plotnikov et al., 1999*; *Plotnikov et al., 2000*; *Schlessinger et al., 2000*).

Starting from the original observation of PI(4,5)P$_2$-dependent oligomerization to involve intermolecular disulfide bridges (*Müller et al., 2015*), the current study resolved the amino acid code of this phenomenon, demonstrating an exclusive role for C95. Nevertheless, as revealed in this study, a substitution of C77 to alanine caused FGF2 secretion from cells to become less efficient (*Figure 1*). This phenomenon could be explained with the proximity of C77 to K54 and K60 on the molecular surface of FGF2, residues that previously have been demonstrated to be part of the molecular interaction interface between FGF2 and the α1 subunit of the Na,K-ATPase (*Legrand et al., 2020*). Indeed, using BLI to study the kinetics of FGF2/α1 interactions, we revealed a C77A substitution alone to severely weaken this interface. Triple substitutions (K54/60E-C77A) caused a complete failure of FGF2 to bind to the α1 subunit of the Na,K-ATPase. With these findings, we could resolve a long-standing question regarding the molecular mechanism of FGF2 membrane translocation, the differential roles of C77 and C95 with regard to FGF2 interactions with α1 and disulfide-mediated dimerization of FGF2 on membrane surfaces. Our observations have important implications for future studies revealing the mechanisms by which higher FGF2 oligomers can trigger and become accommodated in lipidic membrane pores, based on the flexible hinge produced by the C95-C95 disulfide bridge as predicted by MD simulations in this study.

It is important to note that only a few examples exist for the formation of disulfide bridges in soluble protein complexes that reside in the cytoplasm, an observation that has been attributed to the reducing properties of this compartment. For instance, viruses maturing in the cytoplasm can form stable structural disulfide bonds in their coat proteins (*Locker and Griffiths, 1999*; *Hakim and Fass, 2010*). Moreover, a number of cytosolic proteins, including phosphatases, kinases, and transcriptions factors, have been recognized to be regulated by thiol oxidation and disulfide bond formation, formed as a post-translational modification (*Lennicke and Cochemé, 2021*). In numerous cases with direct relevance for our studies on FGF2, disulfide bond formation and other forms of thiol oxidation occur in association with membrane surfaces. In fact, many of these processes are linked to the inner plasma membrane leaflet (*Nordzieke and Medraño-Fernandez, 2018*). Growth factors, hormones, and antigen receptors are observed to activate transmembrane NADPH oxidases generating O$_2^-$/H$_2$O$_2$ (*Brown and Griendling, 2009*). For example, the local and transient oxidative inactivation of membrane-associated phosphatases (e.g. PTEN) serves to enhance receptor associated kinase signaling (*Netto and Machado, 2022*). It is therefore conceivable that similar processes introduce disulfide bridges into FGF2 while assembling into oligomers at the inner plasma membrane leaflet, providing a mechanism that ensures that FGF2 monomers do not prematurely oligomerize in the cytoplasm of cells. In future experiments, it will be important to study disulfide-dependent FGF2 oligomerization in a cellular context, revealing the oxidant and the enzymatic systems that mediate this process at a time scale compatible with the kinetics observed for FGF2 membrane translocation in intact cells (*Dimou et al., 2019*).

In conclusion, with the current study, important insights were obtained regarding the highly dynamic process of PI(4,5)P$_2$-dependent FGF2 oligomerization on membrane surfaces, the key trigger of the central step of the unconventional secretory pathway of FGF2, the remodeling of the plasma membrane lipid bilayer into a lipidic membrane pore with a toroidal architecture. With a number of other cargo proteins secreted by UPS type I pathways in a PI(4,5)P$_2$-dependent manner such as HIV-Tat, Tau, and homeoproteins (*Rayne et al., 2010*; *Debaisieux et al., 2012*; *Rabouille, 2017*; *Katsinelos et al., 2018*; *Merezhko et al., 2018*; *Merezhko et al., 2020*; *Joliot and Prochiantz, 2022*; *Sparn et al., 2022b*), our findings pave the way for a deeper understanding of the general principles of UPS type I pathways of UPS.

## Materials and methods
### Cell culture
CHO, CHO K1, and HeLa S3 cells were cultured in minimal essential medium Eagle (α-MEM) and Dulbecco's modified Eagle's medium, respectively (*Legrand et al., 2020*; *Lolicato et al., 2022*; *Sparn et al., 2022a*). Both media were supplemented with 10% heat-inactivated fetal calf serum, 100 U/mL

penicillin, and 100 µg/mL streptomycin. α-MEM was further supplemented with 2 mM glutamine. Cell lines were grown at 37°C, in the presence of 5% $CO_2$, and with 95% humidity. All parental cell lines used in this study were received from the Leibniz Institute DSMZ (German collection of microorganisms and cell cultures GmbH). For human cell lines, their identities were confirmed by STR profiling. CHO K1 cells' identity and purity were analyzed by a multiplex cell contamination test. All cell lines tested negative for mycoplasma contaminations.

## Generation of stable cell lines

Stable cell lines (CHO and CHO K1) were generated with a retroviral transduction system based on Moloney murine leukemia virus as previously described (*Engling et al., 2002*). Virus production was performed in HEK293 cells with a stably integrated pVPack-Eco packaging system in its genome as well as the retroviral packaging proteins (EcoPack 2-293 cells). Proteins to be expressed upon induction with doxycycline (like FGF2-GFP) were cloned into the pRevTre2 vector, containing a Tet-response element. Retrovirus production was performed according to the MBS Mammalian Transfection Kit (Agilent Technologies) and virus was harvested after 2 days from confluent cells. CHO and CHO K1 cells constitutively expressing the murine cationic amino acid transporter MCAT-1 (*Albritton et al., 1989*) and a Tet-On transactivator, rtTA2-M2 (*Urlinger et al., 2000*), were transduced with the freshly harvested virus. GFP-expressing cells were selected by FACS.

## Gel electrophoresis and western analyses

Samples were loaded on NuPAGE Bis-Tris 4–12% gradient polyacrylamide gels (Thermo Fisher Scientific). SDS-PAGE was conducted in either MES or MOPS SDS running buffer (Thermo Fisher Scientific) at 200 V for 40 min. As a molecular weight marker, a pre-stained PageRuler was used (Thermo Fisher Scientific). Proteins were then transferred to a methanol-activated PVDF membrane (Millipore) for 1 hr at 100 V, in blot buffer (40 mM glycine, 25 mM Tris base, and 20% methanol). After protein transfer, PVDF membranes were blocked for 1 hr at room temperature with PBS containing 5% milk. After that, two washing steps of 5 min each were performed in PBS-T (PBS supplemented with 0.05% Tween-20) at room temperature prior to the addition of a primary antibody solution (in PBS-T supplemented with 2% BSA and 0.02% Na-azide). After 1 hr incubation at room temperature, blots were washed four times with PBS-T prior to the addition of fluorescently labeled secondary antibodies (goat anti-mouse AlexaFluor 680 [Thermo Fisher Scientific] and goat anti-rabbit IRDye 800CW [LI-COR Biosciences]). After 30 min incubation at room temperature, blots were washed four times with PBS-T and once with PBS. Imaging was performed with the LI-COR Odyssey CLx system and analyzed with Image Studio Lite (version 5.0.21, LI-COR Biosciences).

## Cell surface biotinylation

Cell surface biotinylation assays were conducted with stable cell lines producing FGF2 fusion proteins in a doxyclinbe-dependent manner as described previously (*Müller et al., 2015*; *Legrand et al., 2020*; *Sparn et al., 2022a*). $2×10^5$ CHO cells were seeded on six-well plates 48 hr prior the conductance of the assay. These cell lines expressed either WT or mutant forms of FGF2-GFP (C77A, C95A, C77/95A) in a doxycycline-dependent manner. Protein expression was induced with the addition of 1 µg/mL doxycycline 24 hr after seeding. To perform the biotinylation assay, cells were put on ice and washed twice with PBS supplemented with 1 mM $MgCl_2$ and 0.1 mM $CaCl_2$. Subsequently, cells were incubated on ice with 1 mg/mL sulfo-NHS-SS-biotin dissolved in incubation buffer (150 mM NaCl, 10 mM triethanolamine, pH 9.0, 2 mM $CaCl_2$) for 30 min. Biotinylation was stopped by washing the cells once with quenching buffer (100 mM glycine in PBS supplemented with 1 mM $MgCl_2$ and 0.1 mM $CaCl_2$) and incubating for an additional 20 min on ice with quenching buffer. Cells were then washed twice with PBS and lysed for 10 min at 37°C in lysis buffer (62.5 mM EDTA pH 8.0, 50 mM Tris-HCl pH 7.5, 0.4% sodium deoxycholate, 1% Nonidet P-40, and protease inhibitor cocktail [Roche]). Lysed cells were detached by scraping, transferred to fresh 1.5 mL tubes, and sonicated in a sonication bath for 3 min. Cell lysates were then incubated for 15 min at room temperature and vortexed every 5 min. Lysates were centrifuged at 18,000 × $g$ for 10 min at 4°C to remove cell debris. 5% of the supernatant (15 µL out of 300 µL) was used as a total cell lysate input and boiled at 95°C for 10 min after the addition of SDS sample buffer (40% glycerol, 240 mM Tris-HCl pH 6.8, 8% SDS, 5% β-mercaptoethanol, and bromophenol blue). The remaining 95% of whole cell lysates were incubated using continuous

overhead turning at room temperature with Pierce Streptavidin UltraLink Resin (Thermo Fisher Scientific) beads that were previously washed twice with lysis buffer. Washing steps were performed at 3000 × g for 1 min. After 1 hr incubation, unbound material was removed via centrifugation (1 min at 3000 × g, room temperature) and beads were washed once with washing buffer 1 (lysis buffer supplemented with 0.5 M NaCl) and three times with washing buffer 2 (lysis buffer containing 0.1% NP-40, supplemented with 0.5 NaCl). Bound material was then eluted via boiling at 95°C for 10 min with SDS sample buffer. Input and eluate samples were then separated via SDS-PAGE and analyzed by western blot analysis. Western analysis was conducted for FGF2-GFP (detected with an anti-GFP polyclonal rabbit antibody [Custom made, Pineda Antibody Service]) and GAPDH (detected with a mouse monoclonal antibody [Thermo Fisher Scientific]).

### Real-time single molecule TIRF analyses in cells

Real-time single molecule TIRF assays to quantify FGF2 recruitment at the inner plasma membrane leaflet under various experimental conditions were conducted as described previously (*Dimou et al., 2019*; *Legrand et al., 2020*; *Lolicato et al., 2022*). CHO K1 expressing FGF2 variants (WT, C77A, C95A, or C77/95A) as GFP fusion proteins or GFP alone in a doxycycline-dependent manner were cultivated on eight-well glass bottom μ-slides (ibidi) for 24 hr before experiments were started. FGF2-GFP expression levels were kept low in the absence of doxycycline to allow for single molecule measurements. Before imaging, cells were washed twice with Live Cell Imaging Solution (Thermo Fisher Scientific). Both wide-field and TIRF imaging were performed using an inverted Olympus IX81 xCellence TIRF microscope equipped with an Olympus PLAPO 100×1.45 NA Oil DIC objective lens and a Hamamatsu ImagEM Enhanced (C9100-13) camera. Wide-field images were exploited to detect the frames of each cell, and GFP fluorescence was excited with an MT 20 illumination system. TIRF time-lapse videos were analyzed to detect single FGF2-GFP particles recruited at the inner plasma membrane leaflet, and GFP fluorescence was excited with an Olympus 488 nm, 100 mW diode laser. Both wide-field and TIRF images were recorded with the Olympus xCellence software, saved as Tagged Image File Format (TIFF), and analyzed via Fiji (*Schindelin et al., 2012*). FGF2-GFP recruitment efficiencies were quantified after normalizing for both FGF2-GFP expression levels (quantified for each analyzed cell at the first frame of TIRF time-lapse videos) and surface area ($\mu m^2$). To quantify single FGF2-GFP particles recruited at the inner plasma membrane leaflet, the Fiji plugin TrackMate was employed (*Tinevez et al., 2017*). For every representative image shown, background fluorescence was subtracted.

### Key resources table

| Reagent type (species) or resource | Designation | Source or reference | Identifiers | Additional information |
|---|---|---|---|---|
| Gene (*Homo sapiens*) | FGF2, fibroblast growth factor 2 | GenBank | Gene ID: 2247 | NP_001348594.1 18 kDa isoform |
| Gene (*Homo sapiens*) | ATP1A1, ATPase Na+/K+ transporting subunit alpha 1 | GenBank | Gene ID: 476 | |
| Strain, strain background (*Escherichia coli*) | W3110Z1 | *Lutz and Bujard, 1997* | | Chemically competent |
| Strain, strain background (*Escherichia coli*) | BL21 Star (DE3) | Thermo Fisher | | Chemically competent |
| Cell line (Chinese Hamster) | CHO K1 FGF2-GFP (WT) | *Legrand et al., 2020* | | Parental Cell line CHO K1 MCAT Tam 2: *Zehe et al., 2006* |
| Cell line (Chinese Hamster) | CHO K1 FGF2-GFP (C77A) | This work | | Parental Cell line CHO K1 MCAT Tam 2: *Zehe et al., 2006* |
| Cell line (Chinese Hamster) | CHO K1 FGF2-GFP (C95A) | This work | | Parental Cell line CHO K1 MCAT Tam 2: *Zehe et al., 2006* |
| Cell line (Chinese Hamster) | CHO K1 FGF2-GFP (C77/95A) | *Legrand et al., 2020* | | Parental Cell line CHO K1 MCAT Tam 2: *Zehe et al., 2006* |
| Cell line (Chinese Hamster) | CHO K1 GFP | *Legrand et al., 2020* | | Parental Cell line CHO K1 MCAT Tam 2: *Zehe et al., 2006* |

*Continued on next page*

*Continued*

| Reagent type (species) or resource | Designation | Source or reference | Identifiers | Additional information |
|---|---|---|---|---|
| Cell line (Chinese Hamster) | CHO FGF2-GFP (WT) | *Müller et al., 2015* | | Parental Cell line CHO MCAT Tam 2: *Engling et al., 2002* |
| Cell line (Chinese Hamster) | CHO FGF2-GFP (C77A) | *Müller et al., 2015* | | Parental Cell line CHO MCAT Tam 2: *Engling et al., 2002* |
| Cell line (Chinese Hamster) | CHO FGF2-GFP (C95A) | *Müller et al., 2015* | | Parental Cell line CHO MCAT Tam 2: *Engling et al., 2002* |
| Cell line (Chinese Hamster) | CHO FGF2-GFP (C77/95A) | *Müller et al., 2015* | | Parental Cell line CHO MCAT Tam 2: *Engling et al., 2002* |
| Cell line (Chinese Hamster) | CHO FGF2-GFP (C77S) | This work | | Parental Cell line CHO MCAT Tam 2: *Engling et al., 2002* |
| Cell line (Chinese Hamster) | CHO FGF2-GFP (C95S) | This work | | Parental Cell line CHO MCAT Tam 2: *Engling et al., 2002* |
| Cell line (Chinese Hamster) | CHO FGF2-GFP (C77/95S) | This work | | Parental Cell line CHO MCAT Tam 2: *Engling et al., 2002* |
| Cell line (*Homo sapiens*) | Hela S3 FGF2-P2A-GFP (WT) | This work | | Parental Cell line Hela S3: *Sparn et al., 2022b* |
| Cell line (*Homo sapiens*) | Hela S3 FGF2-P2A-GFP (C77A) | This work | | Parental Cell line Hela S3: *Sparn et al., 2022b* |
| Cell line (*Homo sapiens*) | Hela S3 FGF2-P2A-GFP (C95A) | This work | | Parental Cell line Hela S3: *Sparn et al., 2022b* |
| Cell line (*Homo sapiens*) | Hela S3 FGF2-P2A-GFP (C77/C95A) | This work | | Parental Cell line Hela S3: *Sparn et al., 2022b* |
| Antibody | Anti-GFP (rabbit polyclonal) | Custom-made, Pineda Antibody Service | | Dilution (1:500) |
| Antibody | Anti-GAPDH (mouse monoclonal) | Thermo Fisher Scientific | AM4300 | Dilution (1:20,000) |
| Antibody | Anti-rabbit – Secondary Antibody conjugated to IRDye 800CW (goat polyclonal) | LI-COR Biosciences | 926-32211 | Dilution (1:10,000) |
| Antibody | Anti-mouse – Secondary Antibody conjugated to Alexa Fluor 680 (goat polyclonal) | Thermo Fisher Scientific | A21057 | Dilution (1:10,000) |
| Recombinant DNA reagent | pEVOL-pCMF | *Young et al., 2010* | | |
| Recombinant DNA reagent | pET15b-Hisα1-subCD3-WT | *Legrand et al., 2020* | | |
| Recombinant DNA reagent | pET15b-HisFGF2-Y81pCMF-GFP | *Steringer et al., 2017* | | |
| Recombinant DNA reagent | pQE30-HisFGF2 | *Steringer et al., 2012* | | |
| Recombinant DNA reagent | pQE30-HisFGF2-Y81pCMF | *Müller et al., 2015* | | |
| Commercial assay or kit | EZ-Link NHS-PEG4-Biotin | Thermo Scientific | A39259 | |
| Commercial assay or kit | Zeba Spin Desalting Columns | Thermo Scientific | 89882 | |
| Commercial assay or kit | Streptavidin sensors | Sartorius | | SA biosensors, 18-5019 |
| Chemical compound, drug | p-Carboxylmethylphenylalanine (pCMF) | ENAMINE Ltd, Kiev, Ukraine | | |
| Chemical compound, drug | Atto-633 labeled dioleoyl-PE [Atto-633-DOPE] | ATTO-TEC | | |

*Continued on next page*

*Continued*

| Reagent type (species) or resource | Designation | Source or reference | Identifiers | Additional information |
|---|---|---|---|---|
| Software, algorithm | GraphPad Prism, version 9.4 | GraphPad Prism | | |
| Software, algorithm | GROMACS, version 2022 | GROMACS | | |
| Software, algorithm | Data Analysis HT 12.0 | Sartorius | | |
| Other | Lipid Extract Bovine liver PC | Avanti Polar Lipids | 840055 | Powder |
| Other | Lipid Extract Bovine liver PE | Avanti Polar Lipids | 840026 | Powder |
| Other | Lipid Extract Porcine brain PS | Avanti Polar Lipids | 840032 | Powder |
| Other | Lipid Extract Bovine liver PI | Avanti Polar Lipids | 840042 | Powder |
| Other | Lipid Extract Porcine brain [PI(4,5)P$_2$] | Avanti Polar Lipids | 840046 | Powder |
| Other | Lipid Extract Ovine wool cholesterol | Avanti Polar Lipids | 700000 | Powder |
| Other | Lipid Extract Chicken egg SM | Avanti Polar Lipids | 860061 | Powder |
| Other | Synthetic Lipid 16:0 Lissamine Rhod-PE | Avanti Polar Lipids | 810158 | Powder |
| Other | Synthetic Lipid 18:1 Biotinyl-PE | Avanti Polar Lipids | 870282 | Powder |
| Other | Synthetic Lipid 18:1 DGS-NTA [Ni-lipid] | Avanti Polar Lipids | 790404 | Powder |
| Other | Sartorius OctetRed96e instrument | Sartorius | | |

## Transient expression of FGF2 variant forms and cross-linking experiments in HeLa S3 cells

HeLa S3 cells were seeded on six-well plates and transfected after 24 hr of incubation with a plasmid based on pcDNA 3.1 encoding an FGF2-P2A-GFP fusion protein, using the FuGENE transfection reagent (Promega). Following 48 hr of further incubation, whole cell lysates were prepared in a detergent-containing buffer (50 mM HEPES [pH 7.4], 1% Nonidet P-40, 0.25% sodium deoxycholate, 50 mM NaCl, 10% Glycerol, Halt Protease and Phosphatase Inhibitor Cocktail [Thermo Fisher]). Lysates were incubated on ice for 30 min followed by centrifugation at 20,000 × $g$ for 10 min at 4°C. Supernatants were subjected to cross-linking reactions using (i) BMH, (ii) PMPI, or (iii) BMOE that were prepared in DMSO and further diluted to a final concentration of 0.2 mM. Following incubation for 30 min at room temperature in the dark, excess amounts of cross-linkers were quenched for 15 min with 50 mM dithiothreitol (DTT). FGF2 cross-linking products were separated by SDS-PAGE and quantified by western analysis using anti-FGF2 polyclonal rabbit antibody (Custom made, Pineda Antibody Service) and goat anti-rabbit IRDye 800CW (LI-COR Biosciences) as secondary antibody (*Engling et al., 2002*; *Backhaus et al., 2004*; *Schäfer et al., 2004*). The ratio of monomeric versus dimeric species of FGF2 was quantified using a LI-COR Odyssey imaging system. The corresponding data were statistically evaluated as explained in the legend to *Figure 3*.

## Recombinant proteins

His-tagged variants of FGF2 (WT, C77A, C95A, C77/95A, K54/60A, K54/60A-C77A [Biolayer Interferometry]), FGF2-Y81pCMF (WT, C77A, C95A, C77/95A [Membrane Pore Formation Assay]) (both pQE30), FGF2-Y81pCMF-GFP ([WT, C77A, C95A, C77/95A] [Dual-color FCS Measurement and Translocation Assay]), as well as Hisα1-subCD3-WT (Biolayer Interferometry) (both pET15b) were expressed in *Escherichia coli* strains W3110Z1 or BL21 Star (DE3), respectively. For incorporation of the unnatural amino acid *p*-carboxylmethylphenylalanine (pCMF; custom synthesis by ENAMINE Ltd., Kiev, Ukraine), codon 81 (tyrosine) was replaced by an amber stop codon. Transformation of a strain carrying the

pEVOL-pCMF plasmid resulted in expression of recombinant FGF2-Y81pCMF (*Young et al., 2010*). Recombinant proteins were purified to homogeneity in three steps using Ni-NTA affinity chromatography, heparin chromatography (except ATP1A1subCD3), and size exclusion chromatography using a Superdex 75 column (*Steringer et al., 2017*). Protein purity was determined by SDS-PAGE under reducing conditions. For each protein, 3 µg were loaded. Protein patterns were analyzed by Instant Blue Coomassie staining (abcam).

## Lipids

Membrane lipids from natural extracts (bovine liver phosphatidylcholine [PC], bovine liver phosphatidylethanolamine [PE], porcine brain phosphatidylserine [PS], bovine liver phosphatidylinositol [PI], porcine brain PI(4,5)$P_2$, ovine wool cholesterol [Chol], and chicken egg sphingomyelin [SM]) as well as synthetic products (16:0 Lissamine Rhod.-PE, 18:1 Biotinyl-PE [Biotinyl-PE], and 18:1 DGS-NTA (Ni) [Ni-lipid]) were purchased from Avanti Polar Lipids. In addition, Atto-633-labeled dioleoyl-PE [Atto-633-DOPE] was purchased from ATTO-TEC.

## Preparation of GUVs

GUVs with a plasma membrane-like lipid composition consisting of 30 mol% Chol, 15 mol% SM, 34 mol% PC, 10 mol% PE, 5 mol% PS, 5 mol% PI, and 1 mol% Biotinyl-PE (Avanti Polar Lipids) were generated based on electro-swelling using platinum electrodes (*García-Sáez et al., 2009*). GUVs were supplemented with either PI(4,5)$P_2$ or a Ni-NTA lipid at 2 mol% at the expense of PC as indicated. For visualization either 0.05 mol% rhodamine B-labeled PE for FGF2 translocation assays or 0.002 mol% Atto-633-labeled dioleolyl-PE (Atto-633-DOPE, ATTO-TEC) for Dual-color FCS measurements was added. The dried lipid film was hydrated with a 300 mM sucrose solution (300 mOsm/kg, Wescor Vapro). Where indicated, long-chain heparins (50 mM; based on disaccharide units) were included in the lumen of GUVs in order to mimic heparan sulfates. Swelling was conducted at 45°C (10 Hz, 1.5 V for 50 min [without heparin] or 70 min [with heparin], 2 Hz, 1.5 V for 25 min). In order to remove excess amounts of heparin and sucrose, GUVs were gently washed twice with buffer B (25 mM HEPES pH 7.4, 150 mM NaCl, 310 mOsmol/kg) and collected via centrifugation (1200 × *g*; 25°C; 5 min). Imaging chambers (LabTek for FGF2 translocation assays, ibidi for Dual-color FCS) were incubated sequentially with 0.1 mg/mL Biotin-BSA (Sigma A8549) and 0.1 mg/mL Neutravidin (Thermo Fisher Scientific A2666) in buffer B.

## Dual-color FCS measurement

The oligomeric state of His-tagged FGF2-Y81pCMF-GFP variants (WT, C77A, C95A, and C77/95A) was determined by z-scan FCS measurements performed on a home-built confocal microscope system consisting of an Olympus IX71 inverted microscope body (Olympus, Hamburg, Germany) with a 3D piezo positioner from Physik Instrumente (P-562.3CD stage controlled via E-710.3CD controller) and pulsed diode laser heads PicoTA-532, LDH-P-C-470, and LDH-P-635 controlled via PDL 828 Sepia II laser driver (all devices from PicoQuant, Berlin, Germany) at room temperature. The lasers were pulsing alternately in order to avoid artifacts caused by signal bleed-through. The beam was coupled to a single polarization maintaining single mode fiber, collimated by an air space objective (UPlanSApo 4×, NA 0.16) and directed toward the water immersion objective (UPlanSApo 60× w, NA 1.2) by a Chroma ZT375/473/532/635rpc quad-band dichroic mirror. The collected signal, which passed through a 50 µm diameter hole in the focal plane, was split between two SPAD detectors ($PD-50-CTC, MicroPhoton-Devices, Bolzano, Italy) using T635lpxr splitter and HQ515/50 (FGF2-GFP) and HQ697/58 (DOPE-Atto 633) filters mounted in front of each detector. Time-tagged time-resolved single photon counting data acquisition was performed by HydraHarp400 Multichannel Picosecond Event Timer & TCSPC Module controlled via SymPhoTime software (both from PicoQuant, Berlin, Germany). The laser intensity at the back aperture of the objective was kept below 10 mW for each laser line. The z-scan was performed on the top of a selected GUV. The GUV was positioned into the laser beam waist of 470 and 635 nm lasers and moved 1.5 µm below the waist and consequently vertically scanned in 10–15 steps (150 nm spaced). At every position, a 60-s-long dual-color FCS measurement was performed.

## Determination of the FGF2 oligomeric state on a single GUV

The oligomeric state of variant forms of FGF2-Y81pCMF-GFP (WT, C77A, C95A, and C77/95A) was assessed by comparing the brightness of the protein oligomer with the brightness of a defined

monomer. The auto-correlation curves obtained from FCS measurements were fitted by a model assuming 2D diffusion in the membrane (bound FGF2-GFP and DOPE-Atto-633), free 3D diffusion in the solution (FGF2-GFP in the bulk) and transition of the dye to the triplet state (*Widengren et al., 1995*):

$$G\left(\tau\right) = 1 + \left(\frac{1}{N}\frac{1}{1 + \left(\frac{\tau}{\tau_D}\right)} + \frac{1}{N_{\text{free}}}\frac{1}{1 + \left(\frac{\tau}{\tau_{D,\text{free}}}\right)\sqrt{1 + SP\frac{\tau}{\tau_{D,\text{free}}}}}\right)\frac{1 - T + T\exp\left(\frac{-\tau}{\tau_T}\right)}{1 - T} \quad (1)$$

The $\tau$ represents the lag-time, N and $N_{\text{free}}$ are the number of membrane-bound and free dye molecules in the confocal volume, $\tau_D$ and $\tau_{D,free}$ the diffusion times of membrane bound and membrane free dye, SP the structure parameter, T the fraction of the dye in the triplet state, and $\tau_T$ the lifetime of the triplet state. The auto-correlation curves were recorded for both DOPE-Atto633, reporting on the quality of the membrane, and FGF2-GFP that were analyzed to determine the brightness of an FGF2 oligomer. In the beam center, the fluorescent signal coming from the solution is negligible. This simplifies the above equation into:

$$G\left(\tau\right) = 1 + \frac{1}{N}\frac{1}{1 + \left(\frac{\tau}{\tau_D}\right)}\frac{1 - T + T\exp\left(\frac{-\tau}{\tau_T}\right)}{1 - T} \quad (2)$$

Obtaining reliable output parameters requires a precise focus into the beam center, which is achieved by successive vertical scanning of the membrane along the z-axis. For further analysis, the position of the membrane with the minimum in N and the corresponding average intensity in counts per second $\langle I \rangle$ was used (*Benda et al., 2003*). The average oligomeric state of FGF2 on individual GUV was calculated by comparing the brightness of an oligomer $\phi\left(\text{oligo}\right)$ to that of a monomer $\phi\left(\text{mono}\right)$. The brightness of an oligomer was calculated as $\phi\left(\text{oligo}\right) = \frac{\langle I \rangle}{N}$. The brightness of a monomer $\phi\left(\text{mono}\right)$ is obtained in a similar way, however, the presence of one labeled molecule of FGF2-Y81pCM-GFP in a cluster must be ensured. To fulfill this requirement, His-FGF2-Y81pCMF-C77/95A-GFP was diluted by its unlabeled variant (His-FGF2-Y81pCMF-C77/95A) at a ratio of 1:10. Alternatively, recombinant His-FGF2-Y81pCMF-C77/95A-GFP which binds to DGS-NTA containing lipid bilayers as a dimer at maximum was used (*Müller et al., 2015*; *Steringer et al., 2017*; *Šachl et al., 2020*). Finally, the average oligomeric state was calculated as:

$$N\left(\text{m.u.}\right) = \frac{\phi\left(\text{oligo}\right)}{\phi\left(\text{mono}\right)} \quad (3)$$

## Imaging and quantification of FGF2 membrane translocation using GUVs

For FGF2 membrane translocation assays, GUVs were incubated for 3 hr with a small fluorescent tracer (Alexa647) and His-tagged FGF2-Y81pCMF-GFP variants (WT, C77A, C95A, C77/95A) at a final concentration of 200 nM in buffer B (25 mM HEPES pH 7.4, 150 mM NaCl, 310 mOsmol/kg) as described previously (*Steringer et al., 2017*). Confocal images were recorded at room temperature in multitrack mode using Zeiss LSM510 confocal fluorescence microscopes (Carl Zeiss AG, Oberkochen, Germany) using a plan apochromat 63×, NA 1.4 oil immersion objective. Pinholes of the tracks were optimized to 1.2 mm. In order to measure (i) GFP-, (ii) Rhod.-PE-, and (iii) Alexa647-derived signals, samples were excited with (i) an argon laser (488 nm), (ii) a He-Ne-laser (561 nm), or (iii) a He-Ne laser (633 nm). The emission signal was detected after (i) a band-pass (BP) filter (505–530 nm), (ii) a BP filter (560–615 nm), or (iii) a long-pass filter (>650 nm). Images were recorded in 8-bit grayscale. The luminal fluorescence of individual GUVs was measured and normalized to the fluorescence intensity of the surrounding buffer. For each experimental condition, 20–120 individual GUVs were analyzed using ImageJ software (RRID:SCR_003070). To allow for statistical analysis of membrane pore formation and FGF2-GFP membrane translocation across the population of GUVs, thresholds were defined to classify individual GUVs. When the inside-to-outside fluorescence ratio of the Alexa647 tracer was ≥0.6, GUVs

were classified as vesicles containing membrane pores. Similarly, when the inside-to-outside ratio of GFP fluorescence was ≥1.6, the corresponding GUVs were classified as vesicles where FGF2-GFP membrane translocation into the lumen had occurred. Statistical analyses are based on two-tailed, unpaired t-test using GraphPad Prism, version 9.4.1.

## Preparation of LUVs and membrane pore formation assays

Liposomes with a plasma membrane-like lipid composition consisting of 50 mol% Chol, 12.5 mol% SM, 15.5 mol% PC, 9 mol% PE, 5 mol% PS, 5 mol% PI, 2 mol% of PI(4,5)P$_2$, and 1 mol% Rhod.-PE were prepared as described previously (*Temmerman et al., 2008*; *Temmerman and Nickel, 2009*; *Steringer et al., 2012*; *Müller et al., 2015*). In brief, chloroform-dissolved lipid mixtures were first dried under a gentle nitrogen stream and further dried under vacuum for 1.5 hr to yield a homogeneous lipid film. Lipids were resuspended in buffer A (100 mM KCl, 25 mM HEPES, pH 7.4, 10% [wt/vol] sucrose) supplemented with 100 µM concentration of the membrane-impermeant fluorophore 5 (6)-carboxyfluorescein (Sigma) at 45°C to form liposomes with a final lipid concentration of 8 mM. Liposomes were subjected to 10 freeze/thaw cycles (50°C/liquid nitrogen) and to 21 size extrusion steps (400 nm pore size; Avanti Polar Lipids mini-extruder). Liposome preparations were analyzed by dynamic light scattering (Wyatt), indicating a range of 200–400 nm in diameter. To remove extraluminal 5 (6)- carboxyfluorescein, liposomes were diluted in buffer B (150 mM KCl, 25 mM HEPES, pH 7.4) and collected by centrifugation at 15,000 × *g* for 10 min at 20°C followed by size exclusion chromatography using a PD10 column (GE Healthcare). Importantly, this column was operated in buffer C (150 mM KCl, 25 mM HEPES, pH 7.4, 10% [wt/vol] sucrose, 2% [wt/vol] glucose) that was titrated with glucose to reach iso-osmolality (840 mOsm/kg, Wescor Vapro). After incubation with various forms of His-tagged FGF2 (WT, C77A, C95A, C77/95A) at a final concentration of 2 µM, fluorescence dequenching was measured using a SpectraMax M5 fluorescence plate reader (Molecular Devices). At the end of each experiment, Triton X-100 (0.2% [wt/vol] final concentration) was added to measure maximal dequenching used to normalize data.

## BLI to quantify α1 interaction with FGF2 variants

Measurements were conducted on a Sartorius OctetRed96e instrument using Streptavidin sensors (SA, Sartorius 18-5019). This type of sensor was chosen because the nonspecific binding control showed less than 10% of the corresponding signal, which is a prerequisite for BLI measurements. Hisα1-subCD3-WT (*Legrand et al., 2020*) was biotinylated with EZ-Link NHS-PEG4-Biotin (Thermo Scientific A39259) in a 1:1.5 ratio at 25°C for 30 min. Free Biotin was removed using Zeba Spin Desalting Columns (Thermo Scientific 89882) according to the manufacturer's instruction. In order to find the optimal ligand concentration, a loading scout was performed. In the following experiments 4 µg/mL (150 nM) biotinylated Hisα1-subCD3-WT was loaded for 20 min. Next, FGF2-WT was titrated (1:2 dilution series starting at 1000 nM) in order to determine $K_D$, $k_a$, and $k_d$ values. For comparison of FGF2 variants, a concentration of 1000 nM was selected to make use of the full dynamic range of the assay. Kinetic assay step times were as follows: equilibration (5 min), loading of ligand (20 min), wash (5 min), baseline (3 min), association (10 min), dissociation (10 min). All steps were performed in assay buffer (0.02% [wt/vol] Tween-20, 0.1% [wt/vol] BSA in PBS) at 25°C while shaking (1000 rpm).

Data were analyzed with Data Analysis HT 12.0 software version 12.0.2.59 (Sartorius). In brief, a single referencing setup was used (drift) where a loaded sensor was run in parallel in an assay buffer containing a reference well. Later, the resulting signals were subtracted from all sample well data. According to kinetic measurement guidelines data curves were aligned in Y using the function 'Average Baseline Step', inter-step correction 'Dissociation Step', and Savitzky-Golay filter. $K_D$, $k_a$, and $k_d$ values were calculated for FGF2-WT to Hisα1-subCD3-WT by globally fitting Association and Dissociation using a 1:1 model. Curves that showed a high residual were excluded according to the manufacturer's guidelines. Mean with standard deviations of $K_D$, $k_a$, and $k_d$ values (n=3) are given.

## 360° Analysis: sampling of the dimerization interface through atomistic MD simulations

To obtain a coverage of possible FGF2-FGF2 dimerization interfaces compatible with a membrane-bound state, we performed 360 atomistic MD simulations. The goal was to find out all dimerization interfaces where C95 is involved. Both FGF2 monomers were placed on the surface of a model

membrane composed of POPC enriched with 30 mol% Chol. Initially, both monomers pointed in the same direction, based on our previous work where we observed a high-affinity orientation for FGF2 characterized by strong PI(4,5)P$_2$-mediated binding to the membrane surface (*Steringer et al., 2017*, eLife). In this orientation, the known PI(4,5)P$_2$ binding site of FGF2 (K127, R128, K133; *Temmerman et al., 2008*, Traffic) is directly against the membrane surface. We then started with a situation where the C95 residue of the first FGF2 monomer interacted through contact with the surface of the second FGF2 monomer. Next, with the first monomer held in place with its C95 pointed directly toward the second monomer, this second FGF2 monomer was rotated 1° at a time around the z-axis (membrane normal direction), thanks to which we obtained 360 structures in which the FGF2 monomers were in different orientations with respect to each other, and we were able to create initial structures to elucidate all dimerization interfaces where C95 is involved. In each of these cases, the monomers were placed 0.8 nm apart so that in the chosen dimer structure they hit the van der Waals and Coulomb cutoff of the force field (1.2 nm), which ensured the proximity of the monomers but without a strongly bound initial state. For each of the 360 cases, a 500 ns simulation was performed with the previously defined settings. The simulation data was analyzed using machine learning tools. Finally, it is worth noting that checking all possible dimerization interfaces was not the goal of this work, because its implementation with the same resolution would not have been feasible – it would have required 360×360 simulations. Despite this, due to the rotation and diffusion of the protein monomers during the simulations, these simulations also sampled to some extent those dimerization interfaces where C95 was not involved.

## Machine learning-based analysis

To identify the prevalent dimerization interface in the 360° simulations described above, we used the Bayesian Gaussian mixture model (GMM)-based clustering in the structural space spanned by the simulations. The simulation structures were rotationally and translationally fitted to one of the two monomers and its associated PI(4,5)P$_2$ lipid to orient the other monomer in a fixed reference frame. For practical reasons, the dimensionality of the structural space from the fitted trajectory was first reduced using an artificial neural network-based autoencoder (AE) as this workflow allows for more robust clustering. This fitted trajectory was then divided into two equal parts, the training set and the cross-validation set. The training set was used to construct the model weights. The reconstruction error for the cross-validation set in combination with an orthogonal loss was used to train the model over 50 epochs with early stoppings to prevent overfitting (*Wang et al., 2019*). The orthogonal loss function was added to reduce the correlation in the encoded layer neurons. The input layer for the AE was supplied by the coordinates of the C-alpha atoms of the two monomers. The AE was constructed using a total of five dense hidden layers. The two encoding layers were constructed with 1024 neurons followed by the encoded layer with two neurons. These three layers formed in effect the Encoder part of the architecture. The two decoding layers were constructed with 1024 neurons each. The ReLu activation function was used for all neurons. Dropout regularization was used for each layer to enhance the sparsity of the dimensionality reduction model, and L2 regularization was added to avoid over-reliance on highly activating neurons. The pytorch library was used to construct the AE (*Paszke et al., 2019*).

Clusters were identified using Bayesian GMM in the two-dimensional (2D) space encoded by the AE. A 'full' type covariance matrix was used to identify the parameters of the Gaussian distributions used to create the clusters. The initial guesses for the expectation-maximization algorithm for the GMM were placed using the K-means algorithm. Through visual testing, an eight-component model was finally chosen as a robust representation of the clustering in the encoded space. The scikit-learn package was used to train the GMMs (*Pedregosa et al., 2011*).

## In silico protein-protein docking studies

We conducted in silico protein-protein docking studies using the ROSETTA 2018 package (*Gray et al., 2003*; *Wang et al., 2005*; *Wang et al., 2007*; *Chaudhury and Gray, 2008*). Our starting structure was the truncated FGF2 monomer (PDB id: 1BFF; *Kastrup et al., 1997*) from residues 26 to 154 without the flexible N-terminus. We positioned two monomers so that the C95 residues faced each other, with the experimentally known PI(4,5)P$_2$ binding pocket residues oriented toward the same side to allow for simultaneous membrane interactions. To generate diversity, we

randomly perturbed one FGF2 monomer by 3 Å translation and 8° rotation before the start of each docking simulation, resulting in 500 structures. We ranked these structures based on the interface score. The top 5% ranked structures were clustered based on the RMSD value for the FGF2 dimer using the Gromos algorithm (*Daura et al., 1999*). Two structures were considered neighbors if their RMSD value was within 0.6 nm. Finally, we selected the most representative structure from the four-ranked clusters and simulated it in atomistic MD simulations in water as a covalently linked disulfide-bridged dimer.

## Atomistic MD simulations

All simulations were carried out with atomic level models. These MD simulations were performed using the CHARMM36m force field for lipids and proteins, the CHARMM TIP3P force field for water, and the standard CHARMM36 force field for ions (*Huang et al., 2017*). All simulations were carried out using the GROMACS 2022 simulation package (*Abraham et al., 2015*). For FGF2, we used the crystal structure of residues 26–154 of the monomeric form of FGF2 (PDB id: 1BFF; *Kastrup et al., 1997*), with the N- and C-termini modeled as charged residues. All systems were first energy-minimized in a vacuum and then hydrated as well as neutralized by adding an appropriate number of counter ions and 150 mM potassium chloride to mimic experimental conditions. Next, an equilibration step was performed to keep the temperature, pressure, and number of particles constant (NpT ensemble). During this step, proteins were restrained in all dimensions, while the first heavy atom of each lipid (if present) was restrained in the xy-plane of the membrane with a force constant of 1000 kJ/mol/nm$^2$. The Nose-Hoover thermostat maintained the temperature at 310 K with a time constant of 1.0 ps (*Evans and Holian, 1985*). The pressure of 1 atm was kept constant using the Parrinello-Rahman barostat with a time constant set to 5.0 ps and an isothermal compressibility value of $4.5 \times 10^{-5}$ bar$^{-1}$ (*Parrinello and Rahman, 1981*). The isotropic pressure-coupling scheme was used for protein-only simulations, while the semi-isotropic scheme was used in the presence of a membrane. We used the Verlet scheme for neighbor searching with an update frequency of once every 20 steps (*Verlet, 1967*). Electrostatic interactions were calculated using the particle mesh Ewald method (*Darden et al., 1993*) with 0.12 nm spacing, a tolerance of $10^{-5}$, and a cutoff of 1.2 nm. Periodic boundary conditions were applied in all directions. The simulations were carried out using a time step of 2 fs until 1000 ns were reached.

To generate seven C95-C95-dependent FGF2-FGF2 interfaces (protein-only MD simulations), we used three different techniques. First, we extracted one interface from a previous MD simulation snapshot that featured three monomers bound to PI(4,5)P$_2$ on the membrane surface (*Steringer et al., 2017*). Next, we employed ROSETTA software to generate four more initial structures (with the above protocol) from local protein-protein docking simulations. Finally, we obtained the last two interfaces from an AlphaFold2-multimer v3 prediction using ColabFold v1.5.2 default parameters (*Mirdita et al., 2022*). The two C95 residues were then covalently linked to generate the structure and topology files of the seven disulfide bridge dimers, which were then subjected to 1-μs-long MD simulations in water using the CHARMM-GUI web interface (*Lee et al., 2016*). The final structure of one of the previous simulations, which showed a stable dimer in the presence of two ion pairs (E86-K118 and E99-K88), was randomly placed in 10 different orientations 2 nm away from a POPC:Chol (70:30) membrane surface containing two PI(4,5)P$_2$ molecules (protein-membrane MD simulations). These configurations were then simulated for 1 μs.

## Free energy calculations

We employed an atomistic resolution and used the umbrella sampling method (*Torrie and Valleau, 1974*; *Torrie and Valleau, 1977*) to calculate the potential of mean force for the binding of a disulfide-bridged FGF2 dimer on a model membrane surface. The initial configuration for each of the 45 umbrella windows was taken from unbiased MD simulations. The reaction coordinate was set as the center of mass distance between the FGF2 dimer and the phosphate atoms of one leaflet. We simulated each window for 120 ns with a harmonic restraint force constant of 2000 kJ/mol/nm$^2$, spaced 0.1 nm apart. The first 50 ns of the simulations were discarded as the equilibration phase. The weighted histogram analysis method was used to reconstruct the free energy profiles (*Meng and Roux, 2015*). The statistical error was determined with 200 bootstrap analyses (*Rubin, 1981*).

## Chemical cross-linking and peptide purification

For cross-linking, 0.5 mM of freshly resuspended DSBU (Thermo Fisher Scientific, 50 mM stock solution in anhydrous DMSO) was added to 0.5 mg/mL of recombinant His-tagged FGF2 (HEPES-Buffer: 10 mM HEPES, 50 mM NaCl, 1 mM DTT, pH 7.5, 70 µL, pH 7.4–8.2) with and without PI(4,5)P$_2$-liposomes and incubated at room temperature. After 20 min the reaction was quenched by the addition of Tris buffer (1 M, pH 8.0) to a final concentration of 20 mM at room temperature for 30 min. The reaction volume was then doubled by the addition of 100 mM ammonium bicarbonate (ABC) buffer and both crystalline urea and DTT were added to a final concentration 8 M and 5 mM respectively to denature and reduce the cross-linked proteins at 56°C for 30 min. Subsequently, the samples were alkylated by the addition of 8 mM iodoacetamide for 30 min in the dark at room temperature. To start the digestion of the cross-linked proteins, Lys-C (0.25 µg/µL, 2 µL) was added for 4 hr at 37°C. Afterward, the reaction solution was diluted to a final concentration of 2 M urea by the addition 100 mM ABC and the samples were then further digested by the addition of trypsin (1 µg/µL, 2 µL) at 37°C overnight. After digestion, the reaction was stopped by the addition of formic acid to a final concentration of 1% and the resulting peptide mixtures were desalted and purified by in-house-made C18-StageTips (*Rappsilber et al., 2007*), eluted and dried under vacuum.

## Liquid chromatography and XL-MS data analysis

Dried peptides were reconstituted in 10 µL of 0.05% trifluoroacetic acid (TFA), 4% acetonitrile, and 5 µL were analyzed by a Ultimate 3000 reversed-phase capillary nano liquid chromatography (LC) system connected to a Q Exactive HF mass spectrometer (MS) (Thermo Fisher Scientific). Samples were injected and concentrated on a trap column (PepMap100 C18, 3 µm, 100 Å, 75 µm i.d. × 2 cm, Thermo Fisher Scientific) equilibrated with 0.05% TFA in water. After switching the trap column inline, LC separations were performed on a capillary column (Acclaim PepMap100 C18, 2 µm, 100 Å, 75 µm i.d. × 50 cm, Thermo Fisher Scientific) at an eluent flow rate of 300 nL/min. Mobile phase A contained 0.1% formic acid in water, and mobile phase B contained 0.1% formic acid in 80% acetonitrile/20% water. The column was pre-equilibrated with 5% mobile phase B followed by an increase of 5–44% mobile phase B in 130 min. Mass spectra were acquired in a data-dependent mode with a single MS survey scan (375–1575 m/z) with a resolution of 120,000 and MS/MS scans of the 10 most intense precursor ions both with a resolution of 60,000. The dynamic exclusion time was set to 30 s and automatic gain control was set to 3×10$^6$ and 1×10$^5$ for MS and MS/MS scans, respectively. Fragmentation was induced by HCD with a stepped collision energy (21, 27, 33) and MS/MS spectra were recorded using a fixed first mass of 150 m/z. The acquisition of MS/MS spectra was only triggered for peptides with charge states 4+ to 7+.

The acquired RAW files were converted into Peak lists (.mgf format) using Proteome Discoverer (Thermo, version 2.1) containing CID-MS2 data. The CID-MS2 spectra were deconvoluted with the add-on node MS2-Spectrum Processor using default settings. For the main search of the cross-linked peptides, the in-house developed algorithm XlinkX v2.0 was used (*Liu et al., 2017*). The following search parameters were used: MS1 precursor ion mass tolerance: 10 ppm; MS2 fragment ion mass tolerance: 20 ppm; fixed modifications: Cys carbamidomethylation; variable modification: Met oxidation; enzymatic digestion: Trypsin; allowed missed number of missed cleavages, 3. All MS2 spectra were searched against a concatenated target-decoy databases generated based on the WT and mutant sequences of the FGF2 protein. Cross-links were reported at a 2% FDR based on a target decoy calculation strategy (*Liu et al., 2017*). The detected cross-links were mapped on crystal structure PDB (5×10) by setting the allowed distance constraint to the typical range of the used cross-linker (DSBU = ~25 Å).

## Structural modeling of FGF2-Halo dimers into subtomogram average electron density maps

An FGF2-HALO dimer structure was generated that fits well into the 3D density using a structural template of AlphaFold2-multimer (v3) (*Evans et al., 2021*; *Jumper et al., 2021*). All structures generated using AlphaFold2-multimer (v3) are deposited to ZENODO (10.5281/zenodo.10244735). Four of the top-ranked structures resembled a V-shaped dimer, with only the fourth-ranked structure displaying the experimentally known PI(4,5)P$_2$ binding pocket facing toward a membrane-bound state. As the C95 residues were not in direct contact, we replaced the entire FGF2 dimer with the

one obtained from atomistic MD simulations upon binding to the membrane. However, AlphaFold2-multimer (v3) predicted that the two Halo domains were too tightly bound together, which did not align well with the V-shaped density resolved with cryo-ET. To address this, we used the 'Fit in Map' command provided by the ChimeraX software to locally optimize the fit of one of the two Halo domains' atomic coordinates into the density map (*Goddard et al., 2018*; *Pettersen et al., 2021*). As a result, we ensured no clashes with the FGF2 dimer and then applied the exact translation and rotation to the other Halo domain to create a symmetric structure. Finally, we positioned the modeled V-shaped C95-C95 disulfide-bridged FGF2 dimer on the surface of a model membrane (POPC: CHOL [70:30] with 2 PI(4,5)P$_2$ molecules) and subjected it to a 500 ns simulation. We conducted three independent repeats and generated diversity by assigning random initial velocities. The topology of the V-shaped C95 disulfide-bridged dimer was generated using the CHARMM-GUI website.

## Cryo-ET and subtomogram averaging

Proteoliposomes containing FGF2-Y81pCMF-Halo (20 µL) were supplemented with 2.7 µL of Protein A-colloidal gold (10 nm) and applied onto glow discharged Quantifoil grids (R3.5/1). Glow discharging was done using Gatan Solarus 950. Proteoliposomes were vitrified by plunge freezing into liquid ethane using Vitrobot Mark IV (Thermo Fisher Scientific) with the following experimental settings: volume 3.5 µL, temperature 6°C, humidity 100%, blotting force 0, 3 s blotting time. Cryo-ET was performed using the Titan Krios (Thermo Fisher Scientific) equipped with an energy filter and K2 direct electron detector (Gatan). Tilt series were acquired at a magnification of ×64,000 either in-focus using VPP at EMBL, Heidelberg (pixel size 0.213 nm) or at nominal defocus –4 µm without the VPP at Heidelberg University (pixel size 0.229 nm) using an energy filter zero loss peak window set to 20 eV. Mapping and tilt series acquisition was done using SerialEM (*Mastronarde, 2005*) and dose-symmetric tilt series schema with an increment angle 3° and a tilt range of 60° with cumulative electron dose 140 e$^-$/Å$^2$ (*Hagen et al., 2017*). Projection records were collected in counting mode as dose-fractionated movies and aligned on-fly using SerialEM 4 K plug-in. Tilt series alignment with gold fiducial markers, CTF correction (except for VPP data), dose-filtration and tomogram reconstruction were done in IMOD (etomo GUI) using weighted back-projection with SIRT-like filter with seven iterations (*Mastronarde and Held, 2017*). Subtomogram extraction and averaging were performed in Dynamo (*Castaño-Díez et al., 2012*; *Castaño-Díez, 2017*; *Castaño-Díez et al., 2017*; *Navarro et al., 2018*). At first, V-shaped particles were selected (N=186) using a dipole model with a north direction normal to the center of the liposome at different positions with respect to the tilt axis of the tomogram acquired with VPP. Subvolumes with a cubic size of 160 voxels were extracted and averaged to create a template. Particles were iteratively aligned and averaged using a mask focusing on the protein density and a membrane and alternating c1 and c2 symmetry at each iteration. The final subtomogram average was visualized in ChimeraX (*Goddard et al., 2018*; *Pettersen et al., 2021*) where manual docking of Halo domain (PDB:4KAJ) and FGF2 monomer (PDB:1BFF) crystal structures was performed.

## Acknowledgements

This work was supported by the Deutsche Forschungsgemeinschaft (SFB/TRR 186, project A1 and A5; WN and CF, DFG Ni 423/10-1, DFG Ni 423/12-1 and DFG Ni 423/13-1; WN and DFG LO 2821/1-1; FL). RŠ and MH acknowledge GAČR grant 20-01401J. We thank the cryo-EM network at Heidelberg University (HD-cryoNET) for support and assistance and the Electron Microscopy Core Facility at EMBL and Wim Hagen for data acquisition funded by iNext. The authors gratefully acknowledge the data storage service SDS@hd supported by the Ministry of Science, Research, and the Arts Baden-Württemberg (MWK), the German Research Foundation (DFG) through grant INST 35/1314-1 FUGG and INST 35/1503-1 FUGG. We also would like to acknowledge support from Holger Lorenz (ZMBH Imaging facility) and Monika Langlotz (ZMBH FACS Facility). The Vattulainen group has been supported by the Academy of Finland (projects 331349, 336234, 346135), the Sigrid Juselius Foundation, Helsinki Institute of Life Science (HiLIFE) Fellow Program, the Human Frontier Science Program (RGP0059/2019), the Lundbeck Foundation, and DFG (SFB/TRR 83) (IV). This project has received funding from the European Union's Horizon 2020 research and innovation programme under the Marie Skłodowska-Curie grant agreement No. 101033606 (SK). We acknowledge the computing resources provided by the CSC – IT Center for Science Ltd. (Espoo, Finland). We further would like

to thank Tobias P Dick (DKFZ Heidelberg) who provided valuable insights into oxidative processes producing disulfide-bridged protein complexes at the inner plasma membrane leaflet.

## Additional information

### Funding

| Funder | Grant reference number | Author |
| --- | --- | --- |
| Deutsche Forschungsgemeinschaft | DFG LO 2821/1-1 | Fabio Lolicato |
| Deutsche Forschungsgemeinschaft | SFB/TRR 186 project A1 | Walter Nickel |
| Deutsche Forschungsgemeinschaft | DFG Ni 423/10-1 | Walter Nickel |
| Deutsche Forschungsgemeinschaft | DFG Ni 423/12-1 | Walter Nickel |
| Deutsche Forschungsgemeinschaft | DFG Ni 423/13-1 | Walter Nickel |
| Deutsche Forschungsgemeinschaft | SFB/TRR 186 project A5 | Christian Freund |
| Grantová Agentura České Republiky | 20-01401 | Martin Hof |
| Deutsche Forschungsgemeinschaft | INST 35/1314-1 FUGG | Fabio Lolicato |
| Deutsche Forschungsgemeinschaft | INST 35/1503-1 FUGG | Fabio Lolicato |
| Academy of Finland | 331349 | Ilpo Vattulainen |
| Academy of Finland | 336234 | Ilpo Vattulainen |
| Academy of Finland | 346135 | Ilpo Vattulainen |
| Human Frontier Science Program | RGP0059/2019 | Ilpo Vattulainen |
| Lundbeck Foundation | | Ilpo Vattulainen |
| Sigrid Juséliuksen Säätiö | | Ilpo Vattulainen |
| Deutsche Forschungsgemeinschaft | SFB/TRR 83 | Ilpo Vattulainen |
| HORIZON EUROPE Marie Sklodowska-Curie Actions | 101033606 | Shreyas Kaptan |
| CSC – IT Center for Science | | Fabio Lolicato |

The funders had no role in study design, data collection and interpretation, or the decision to submit the work for publication.

### Author contributions

Fabio Lolicato, Conceptualization, Data curation, Formal analysis, Funding acquisition, Investigation, Methodology, Project administration, Resources, Software, Validation, Visualization, Writing – original draft, Writing – review and editing; Julia P Steringer, Conceptualization, Resources, Investigation, Visualization, Methodology, Writing – original draft, Writing – review and editing; Roberto Saleppico, Conceptualization, Resources, Visualization, Methodology; Daniel Beyer, Sebastian Unger, Steffen Klein, Resources; Jaime Fernandez-Sobaberas, Resources, Visualization; Petra Riegerová, Sabine Wegehingel, Hans-Michael Müller, Xiao J Schmitt, Conceptualization, Resources; Shreyas Kaptan, Conceptualization, Software, Resources, Funding acquisition, Validation, Visualization, Methodology,

Writing – review and editing; Christian Freund, Conceptualization, Supervision, Funding acquisition, Methodology, Writing – review and editing; Martin Hof, Supervision, Funding acquisition, Methodology; Radek Šachl, Conceptualization, Resources, Supervision, Funding acquisition, Investigation, Methodology; Petr Chlanda, Conceptualization, Resources, Supervision, Validation, Visualization, Methodology, Writing – review and editing; Ilpo Vattulainen, Data curation, Formal analysis, Funding acquisition, Writing – review and editing; Walter Nickel, Conceptualization, Data curation, Funding acquisition, Investigation, Project administration, Resources, Supervision, Validation, Visualization, Writing – original draft, Writing – review and editing

### Author ORCIDs

Fabio Lolicato (ID) https://orcid.org/0000-0001-7537-0549
Julia P Steringer (ID) https://orcid.org/0000-0001-9418-2762
Roberto Saleppico (ID) http://orcid.org/0000-0003-0502-192X
Hans-Michael Müller (ID) http://orcid.org/0000-0002-2384-7285
Petr Chlanda (ID) http://orcid.org/0000-0002-7782-2139
Ilpo Vattulainen (ID) http://orcid.org/0000-0001-7408-3214
Walter Nickel (ID) https://orcid.org/0000-0002-6496-8286

Reviewer #1 (Public Review): https://doi.org/10.7554/eLife.88579.3.sa1
Reviewer #2 (Public Review): https://doi.org/10.7554/eLife.88579.3.sa2
Reviewer #3 (Public Review): https://doi.org/10.7554/eLife.88579.3.sa3
Author Response https://doi.org/10.7554/eLife.88579.3.sa4

---

## Additional files

### Supplementary files
• MDAR checklist

### Data availability

All structures generated using AlphaFold2-multimer (v3), as well as all initial and configuration molecular dynamics simulations files related to this study, have been deposited to ZENODO. The machine learning code to analyze the MD simulation data has been deposited to GitLab (copy archived at *Kaptan and Vattulainen, 2023*). The source data underpinning the plots presented in this study are available for download as supplementary Excel files.

The following dataset was generated:

| Author(s) | Year | Dataset title | Dataset URL | Database and Identifier |
|---|---|---|---|---|
| Lolicato F | 2023 | Disulfide bridge-dependent dimerization triggers FGF2 membrane translocation into the extracellular space | https://doi.org/10.5281/zenodo.10244735 | Zenodo, 10.5281/zenodo.10244735 |

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
