## [Editor Report · eLife assessment]

This manuscript reports **important** findings, demonstrating a critical role for a cysteine-containing dimerization interface in the secretion of FGF2 through an unconventional pathway. The authors provide **compelling** evidence, combining in vitro biochemical assays with structural simulation. The work will be of interest to researchers working on protein trafficking and secretion.

---

## [Referee Report · Reviewer #1 (Public Review)]

The manuscript by Lolicato and colleagues characterizes the role of FGF2 dimerization in unconventional secretion of this signaling molecule using a combination of cell-based and in vitro assays. FGF2 is a signaling molecule secreted from the cell via an unconventional mechanism because it lacks a signal sequence. Previous studies by the same group have established a compelling model in which FGF2 forms an oligomer in a PIP2 dependent manner at the plasma member, which drives its translocation to the cell exterior. The same group also reports two cysteine residues that are critical for FGF2 oligomerization and secretion.

In this study, the authors analyzed the impact of single Cysteine to Alanine substitution on oligomerization and secretion of FGF2. They found that C95 but not C77 is required for PIP2 dependent membrane binding, FGF2 oligomerization and secretion. On the other hand, C77 is required for the interaction of FGF2 with the plasma membrane Na, K-ATPase, which is thought to enhance the FGF2-PIP2 interaction. Using a set of bi-functional crosslinkers, the authors were able to capture a fraction of the FGF2 homo-dimer, which is dependent on C95. They propose that FGF2 forms a disulfide-bridged dimer via C95, which serves as the building block for FGF2 oligomerization in the plasma membrane.

The revised manuscript has carefully addressed my concerns. I should clarify that when I inquired about evidence for a disulfide-linked FGF2 dimer, I referred to in vivo evidence. I was aware of the authors' previous in vitro study, which demonstrated that FGF2 indeed can form a disulfide dimer under an in vitro condition. Although the new manuscript still contains no in vivo data on this issue, the authors have added numerous controls. In particular, the fact that the FGF2 C95S mutant is severely defective in secretion does provide strong support for the involvement of the thiol group of C95 in FGF2 secretion. The additional discussions on other examples of cytosolically-localized disulfide proteins and those in proximity to membranes further alleviates my concern.

---

## [Referee Report · Reviewer #2 (Public Review)]

Unconventional secretion refers to the release of cargoes without a signal peptide and is performed independent of ER-Golgi trafficking. One essential type of unconventional secretion is type I, in which a cargo can translocate directly across the plasma membrane. FGF2 is one excellent mode to study type I translocation and the authors have focused on FGF2 secretion for decades. Many beautiful works have been performed to reveal the mechanism of FGF2 translocation step by step. And the picture is getting clearer which time a new work from the lab is published. In the current work, the authors characterized the importance of disulfate bond formation on C95 of FGF2 in lipid binding and translocation. In addition, they clearified the role of another C77 which is require for binding to the Na/K -ATPase that regulates the early step of FGF2 binding to the membrane. The authors also employed structural approaches and MD to provide mechanistic insights into the translocation process. In general it is an important advance regarding the translocation of FGF2 and data provided are brief, clear and convincing.

---

## [Referee Report · Reviewer #3 (Public Review)]

In addition to ER-Golgi-dependent conventional protein secretion, a wide range of substrates lacking N-terminal signal peptides are secreted through diverse pathways collectively known as unconventional protein secretion (UPS). The translocation mechanism of these different substrates across the membrane remains a fascinating question in this field. In this manuscript, the authors employ a comprehensive combination of biochemistry, cell biology, and structural biology techniques to investigate the mechanism by which two crucial cystine residues, C77 and C95, facilitate the secretion of FGF2. The key finding is that the C95-C95 disulfide bond mediates the formation of an FGF2 dimer, which is essential for pore formation and translocation. Additionally, it is revealed that C77 promotes FGF2 secretion by interacting with a cell surface factor called Na-K ATPase. This observation provides valuable mechanistic insights into a critical step of FGF2 secretion. Overall, the experimental results presented in this study are both clear and convincing.

The authors have well addressed my concern about the formation of disulfide bond in the revision. In addition, the cross-linking mass spectrometry identified an additional dimerization interface, which would be of interest for future studies on its role in regulating high-order FGF2 oligomer formation and secretion.

---

## [Author Response]

The following is the authors’ response to the original reviews.

**Public Review:**
1. Evidence for a disulfide bridge contained in membrane-associated FGF2 dimers

This aspect was brought up in detail by both Reviewer #1 and Reviewer #3. It has been addressed in the revised manuscript by (i) new experimental and computational analyses, (ii) a more detailed discussion of previous work from our lab in which experiments were done the reviewers were asking for and (iii) a more general discussion of known examples of disulfide formation in protein complexes with a particular focus on membrane surfaces facing the cytoplasm, the inner plasma membrane leaflet being aprominent example. Please find our detailed comments in our direct response to Reviewers #1 and #3, see below.

2. Affinity towards PI(4,5)P2 comparing FGF2 dimers versus monomers

This is an aspect that has been raised by Reviewer 3 along with additional comments on the interaction of FGF2 with PI(4,5)P2. Please find our detailed response below. With regard to PI(4,5)P2 affinity aspects of FGF2 dimers versus FGF2 monomers, we think that the increased avidity of FGF2 dimers with two high affinity binding pockets for PI(4,5)P2 are a good explanation for the different values of free energies of binding that were calculated from the atomistic molecular dynamics simulations shown in Fig. 9. This phenomenon is well known for many biomolecular interactions and is also consistent with the cryoEM data contained in our manuscript, showing a FGF2 dimer with two PI(4,5)P2 binding sites facing the membrane surface.

3. C95-C95 FGF2 dimers as signaling units

We have put forward this hypothesis since in structural studies analyzing the FGF ternary signaling complex consisting of FGF2, FGF receptor and heparin, FGF2 mutants were used that lack C95. Nevertheless, two FGF2 molecules are contained in FGF signaling complexes. In addition to the papers on the structure of the FGF signaling complex, we have cited work that showed that C95-C95 crosslinked FGF2 dimers are efficient FGF signaling modules (Decker et al, 2016; Nawrocka et al, 2020). Therefore, being based on an assembly/disassembly mechanism with the transient formation of poreforming FGF2 oligomers, we think it is an interesting idea that the FGF2 secretion pathway produces C95-C95 disulfide-linked FGF2 dimers at the outer plasma membrane leaflet that can engage in FGF2 ternary signaling complexes. While this is a possibility we put forward to stimulate the field, it of course remains a hypothesis which has been clearly indicated as such in the revised manuscript.

**Reviewer #1:**
1. Evidence for disulfide-bridged FGF2 dimers and higher oligomers on non-reducing versus reducing SDS gels

The experiment suggested by Reviewer #1 is an important one that has been published by our group in previous work. In these studies, we found FGF2 oligomers analyzed on non-reducing SDS gels to be sensitive to DTT, turning the vast majority of oligomeric FGF2 species into monomers [(Müller et al, 2015); Fig. 3, compare panel D with panel H]. This phenomenon could be observed most clearly after short periods of incubations (0.5 hours) of FGF2 with PI(4,5)P2-containing liposomes. These findings constituted the original evidence for PI(4,5)P2-induced FGF2 oligomerization to depend on the formation of intermolecular disulfide bridges.

In the current manuscript, we established the structural principles underlying this process and identified C95 to be the only cysteine residue involved in disulfide formation. Based on biochemical cross-linking experiments in cells, cryo-electron tomography, predictions from AlphaFold-2 Multimer and molecular dynamics simulations, we demonstrated a strong FGF2 dimerization interface in which C95 residues are brought into close proximity when FGF2 is bound to membranes in a PI(4,5)P2-dependent manner. These findings provide the structural basis by which disulfide bridges can be formed from the thiols contained in the side chains of two C95 residues directly facing each other in the dimerization interface. In the revised manuscript, we included additional data that further strengthen this analysis. In the experiments shown in the new Fig. 10, we combined chemical cross-linking with mass spectrometry, further validating the reported FGF2 dimerization interface. In addition, illustrated in the new Fig. 8, we employed a new computational analysis combining 360 individual atomistic molecular dynamics simulations, each spanning 0.5 microseconds, with advanced machine learning techniques. This new data set corroborates our findings, demonstrating that the C95-C95 interface self-assembles independently of C95-C95 disulfide formation, based on electrostatic interactions. Intriguingly, it is consistent with our experimental findings based on cross-linking mass spectrometry (new Fig. 10) where cross-linked peptides could also be observed with the C77/95A variant form of FGF2, suggesting a protein-protein interface whose formation does not depend on disulfide formation. Therefore, we propose that disulfide formation occurs in a subsequent step, representing the committed step of FGF2 membrane translocation with the formation of disulfide-bridged FGF2 dimers being the building blocks for pore-forming FGF2 oligomers.

As a more general remark on the mechanistic principles of disulfide formation in different cellular environments, we would like to emphasize that it is a common misconception that the reducing environment of the cytoplasm generally makes the formation of disulfide bridges unlikely or even impossible. From a biochemical point of view, the formation of disulfide bridges is not limited by a reducing cellular environment but is rather controlled by kinetic parameters when two thiols are brought into proximity. Indeed, it has become well established that disulfide bridges can also be formed in compartments other than the lumen of the ER/Golgi system, including the cytoplasm. For example, viruses maturing in the cytoplasm can form stable structural disulfide bonds in their coat proteins (Locker & Griffiths, 1999; Hakim & Fass, 2010). Moreover, many cytosolic proteins, including phosphatases, kinases and transcriptions factors, are now recognized to be regulated by thiol oxidation and disulfide bond formation, formed as a post-transcriptional modification (Lennicke & Cocheme, 2021). In numerous cases with direct relevance for our studies on FGF2, disulfide bond formation and other forms of thiol oxidation occur in association with membrane surfaces. In fact, many of these processes are linked to the inner plasma membrane leaflet (Nordzieke & Medrano-Fernandez, 2018). Growth factors, hormones and antigen receptors are observed to activate transmembrane NADPH oxidases generating O2·-/H2O2 (Brown & Griendling, 2009). For example, the local and transient oxidative inactivation of membrane-associated phosphatases (e.g., PTEN) serves to enhance receptor associated kinase signaling (Netto & Machado, 2022). It is therefore conceivable that similar processes introduce disulfide bridges into FGF2 while assembling into oligomers at the inner plasma membrane leaflet. In the revised version of our manuscript, we have discussed the above-mentioned aspects in more detail, with the known role of NADPH oxidases in disulfide formation at the inner plasma membrane leaflet being highlighted.

**Reviewer #2:**
1. Potential effects of a C95A substitution on protein folding and comparison with a C95S substitution with regard to phenotypes observed in FGF2 secretion

A valid point that we indeed addressed at the beginning of this project. Most importantly, we tested whether both FGF2 C95A and FGF2 C95S are characterized by severe phenotypes in FGF2 secretion efficiency. As shown in the revised Fig. 1, cysteine substitutions by serine showed very similar FGF2 secretion phenotypes compared to cysteine to alanine substitutions (Fig. 1C and 1D). In addition, in the pilot phase of this project, we also compared recombinant forms of FGF2 C95A and FGF2 C95S in various in vitro assays. For example, we tested the full set of FGF2 variants in membrane integrity assays as the ones contained in Fig. 4. As shown in Author response image 1, FGF2 variant forms carrying a serine in position 95 behaved in a very similar manner as compared to FGF2 C95A variant forms. Relative to FGF2 wild-type, membrane pore formation was strongly reduced for both types of C95 substitutions. By contrast, both FGF2 C77S and C77A did show activities that were similar to FGF2 wild-type.

**Author response image 1. sa4fig1:** 

From these experiments, we conclude that changes in protein structure are not the basis for the phenotypes we report on the C95A substitution in FGF2.

2. Effects of a C77A substitution on FGF2 membrane recruitment in cells

The effect of a C77A substitution in FGF2 recruitment to the inner plasma membrane leaflet is indeed a moderate one. This is likely to be the case because C77 is only one residue of a more complex surface that contacts the α1 subunit of the Na,K-ATPase. Stronger effects can be observed when K54 and K60 are changed, residues that are positioned in close proximity to C77 (Legrand et al, 2020). Nevertheless, as shown in the revised Fig. 1, we consistently observed a reduction in membrane recruitment when comparing FGF2 C77A with FGF2 wild-type. When analyzing the raw data without GFP background subtraction, a significant reduction of FGF2 C77A was observed compared to FGF2 wild-type (Fig. 1A and 1B). We therefore conclude that C77 does not only play a role in FGF2/α1 interactions in biochemical assays using purified components (Fig. 7) but also impairs FGF2/α1 interactions in a cellular context (Fig. 1A and 1B).

3. Identity of the protein band in Fig. 3 labeled with an empty diamond

This is a misunderstanding as we did not assign this band to a FGF2-GFP dimer. When we produced the corresponding cell lines, we used constructs that link FGF2 with GFP via a ‘self-cleaving’ P2A sequence. During translation, even though arranged on one mRNA, this causes the production of FGF2 and GFP as separate proteins in stoichiometric amounts, the latter being used to monitor transfection efficiency. However, a small fraction is always expressed as a complete FGF2-P2A-GFP fusion protein (a monomer). This band can be detected with the FGF2 antibodies used and was labeled in Fig. 3 by an empty diamond.

4. Labeling of subpanels in Fig. 5A

We have revised Fig. 5 according to the suggestion of Reviewer #2.

5. FGF2 membrane binding efficiencies shown in Fig. 5C

It is true that FGF2 variant forms defective in PI(4,5)P2-dependent oligomerization (C95A and C77/95A) bind to membranes with somewhat reduced efficiencies. This is also evident form the intensity profiles shown in Fig. 5A and was observed in biochemical in vitro experiments as well. A plausible explanation for this phenomenon would be the increased avidity when FGF2 oligomerizes, stabilizing membrane interactions (see also Fig. 9B).

6. Residual activities of FGF2 C95A and C77/95A in membrane pore formation?

We do not assign the phenomenon in Fig. 5 Reviewer #2 is referring to as controlled activities of FGF2 C95A and C77/95A in membrane pore formation. Rather, GUVs containing PI(4,5)P2 are relatively labile structures with a certain level of integrity issues upon protein binding and extended incubation times being conceivable. It is basically a technical limitation of this assay with GUVs incubated with proteins for 2 hours. Even after substitution of PI(4,5)P2 with a Ni-NTA membrane lipid, background levels of loss of membrane integrity can be observed (Fig. 6). Therefore, as compared to FGF2 C95A and C77/95A, the critical point here is that FGF2 wt and FGF2 C77A do display significantly higher levels of a loss of membrane integrity in PI(4,5)P2-containing GUVs, a phenomenon that we interpret as controlled membrane pore formation. By contrast, all variant forms of FGF2 show only background levels for loss of membrane integrity in GUVs containing the Ni-NTA lipid.

7. Why does PI(4,5)P2 induce FGF2 dimerization?

This has been studied extensively in previous work (Steringer et al, 2017). As also discussed in the current manuscript, the interaction of FGF2 with membranes through its high affinity PI(4,5)P2 binding pocket orients FGF2 molecules on a 2D surface that increase the likelihood of the formation of the C95containing FGF2 dimerization interface. Moreover, in the presence of cholesterol at levels typical for plasma membranes, PI(4,5)P2 clusters containing up to 4 PI(4,5)P2 molecules (Lolicato et al, 2022), a process that may further facilitate FGF2 dimerization.

8. Is it possible to pinpoint the number of FGF2 subunits in oligomers observed in cryo-electron tomography?

We indeed took advantage of the Halo tags that appear as dark globular structures in cryo-electron tomography. For most FGF2 oligomers with FGF2 subunits on both sides of the membrane, we could observe 4 to 6 Halo tags which is consistent with the functional subunit number that has been analyzed for membrane pore formation (Steringer et al., 2017; Sachl et al, 2020; Singh et al, 2023). However, since the number of higher FGF2 oligomers we observed in cryo-electron tomography was relatively small and the nature of these oligomers appears to be highly dynamic, caution should be taken to avoid overinterpretation of the available data.

**Reviewer #3:**
1. Conclusive demonstration of disulfide-linked FGF2 dimers

A similar point was raised by Reviewer #1, so that we would like to refer to our response on page 2, see above.

2. Identity of FGF2-P2A-GFP observed in Fig. 3

Again, a similar point has been made, in this case by Reviewer #2 (Point 3). The observed band is not a FGF2-P2A-GFP dimer but rather the complete FGF2-P2A-GFP fusion protein (a monomer) that corresponds to a small population produced during mRNA translation where the P2A sequence did not cause the production of FGF2 and GFP as separate proteins in stoichiometric amounts.

3. Quantification of GFP signals in Fig. 6

Fig. 6 has been revised according to the suggestion of Reviewer #3. A comprehensive comparison of PI(4,5)P2 and the Ni-NTA membrane lipid in FGF2 membrane translocation assays is also contained in previous work that introduced the GUV-based FGF2 membrane translocation assay (Steringer et al., 2017).

4. Experimental evidence for various aspects of FGF2 interactions with PI(4,5)P2

Most of the points raised by Reviewer #3 have been addressed in previous work. For example, FGF2 has been demonstrated to dimerize only on membrane surfaces containing PI(4,5)P2 (Müller et al., 2015). In solution, FGF2 remained a monomer even after hours of incubation as analyzed by native gel electrophoresis and reducing vs. non-reducing SDS gels (see Fig. 3 in Müller et al, 2015). In the same paper, the first evidence for a potential role of C95 in FGF2 oligomerization has been reported, however, at the time, our studies were limited to FGF2 C77/95A. In the current manuscript, the in vitro experiments shown in Figs. 2 to 6 establish the unique role of C95 in PI(4,5)P2-dependent FGF2 oligomerization. As discussed above, FGF2 oligomers have been shown to contain disulfide bridges based on analyses on non-reducing gels in the absence and presence of DTT (Müller et al., 2015).

References

Brown DI, Griendling KK (2009) Nox proteins in signal transduction. Free Radic Biol Med 47: 1239-1253Decker CG, Wang Y, Paluck SJ, Shen L, Loo JA, Levine AJ, Miller LS, Maynard HD (2016) Fibroblast growth factor 2 dimer with superagonist in vitro activity improves granulation tissue formation during wound healing. Biomaterials 81: 157-168

Hakim M, Fass D (2010) Cytosolic disulfide bond formation in cells infected with large nucleocytoplasmic DNA viruses. Antioxid Redox Signal 13: 1261-1271

Legrand C, Saleppico R, Sticht J, Lolicato F, Muller HM, Wegehingel S, Dimou E, Steringer JP, Ewers H, Vattulainen I et al (2020) The Na,K-ATPase acts upstream of phosphoinositide PI(4,5)P2 facilitating unconventional secretion of Fibroblast Growth Factor 2. Commun Biol 3: 141

Lennicke C, Cocheme HM (2021) Redox metabolism: ROS as specific molecular regulators of cell signaling and function. Mol Cell 81: 3691-3707

Locker JK, Griffiths G (1999) An unconventional role for cytoplasmic disulfide bonds in vaccinia virus proteins. J Cell Biol 144: 267-279

Lolicato F, Saleppico R, Griffo A, Meyer A, Scollo F, Pokrandt B, Muller HM, Ewers H, Hahl H, Fleury JB et al (2022) Cholesterol promotes clustering of PI(4,5)P2 driving unconventional secretion of FGF2. J Cell Biol 221

Müller HM, Steringer JP, Wegehingel S, Bleicken S, Munster M, Dimou E, Unger S, Weidmann G, Andreas H, GarciaSaez AJ et al (2015) Formation of Disulfide Bridges Drives Oligomerization, Membrane Pore Formation and Translocation of Fibroblast Growth Factor 2 to Cell Surfaces. J Biol Chem 290: 8925-8937

Nawrocka D, Krzyscik MA, Opalinski L, Zakrzewska M, Otlewski J (2020) Stable Fibroblast Growth Factor 2 Dimers with High Pro-Survival and Mitogenic Potential. Int J Mol Sci 21

Netto LES, Machado L (2022) Preferential redox regulation of cysteine-based protein tyrosine phosphatases: structural and biochemical diversity. FEBS J 289: 5480-5504

Nordzieke DE, Medrano-Fernandez I (2018) The Plasma Membrane: A Platform for Intra- and Intercellular Redox Signaling. Antioxidants (Basel) 7

Sachl R, Cujova S, Singh V, Riegerova P, Kapusta P, Muller HM, Steringer JP, Hof M, Nickel W (2020) Functional Assay to Correlate Protein Oligomerization States with Membrane Pore Formation. Anal Chem 92: 14861-14866

Singh V, Macharova S, Riegerova P, Steringer JP, Muller HM, Lolicato F, Nickel W, Hof M, Sachl R (2023) Determining the Functional Oligomeric State of Membrane-Associated Protein Oligomers Forming Membrane Pores on Giant Lipid Vesicles. Anal Chem 95: 8807-8815

Steringer JP, Lange S, Cujova S, Sachl R, Poojari C, Lolicato F, Beutel O, Muller HM, Unger S, Coskun U et al (2017) Key steps in unconventional secretion of fibroblast growth factor 2 reconstituted with purified components. eLife 6: e28985